

# Annual cycle of aerosol properties over the central Arctic during MOSAiC 2019-2020 — light-extinction, CCN, and INP levels from the boundary layer to the tropopause

Albert Ansmann[1], Kevin Ohneiser[1], Ronny Engelmann[1], Martin Radenz[1], Hannes Griesche[1], Julian Hofer[1], Dietrich Althausen[1], Jessie M. Creamean[2], Matthew C. Boyer[3], Daniel A. Knopf[4], Sandro Dahlke[5], Marion Maturilli[5], Henriette Gebauer[1], Johannes Bühl[1], Cristofer Jimenez[1], Patric Seifert[1], and Ulla Wandinger[1]

[1]Leibniz Institute for Tropospheric Research, Leipzig, Germany
[2]Department of Atmospheric Science, Colorado State University, Fort Collins, CO 80526, USA
[3]Institute for Atmospheric and Earth System Research /Physics, University of Helsinki, Helsinki, Finland
[4]School of Marine and Atmospheric Sciences, Stony Brook University, Stony Brook, NY 11794, USA
[5]Alfred Wegener Institute, Helmholtz Centre for Polar and Marine Research, Potsdam, Germany

**Correspondence:** A. Ansmann
(albert@tropos.de)

**Abstract.** Continuous height-resolved observations of aerosol profiles over the central Arctic throughout a full year were performed for the first time. Such measurements covering aerosol layering features are required for an adequate modeling of Arctic climate conditions, especially with respect to a realistic consideration of cloud formation and here, in particular, of ice nucleation processes. MOSAiC (Multidisciplinary drifting Observatory for the Study of Arctic Climate) offered this favorable opportunity to monitor aerosol and clouds over the central Arctic over all four seasons, from October 2019 to September 2020. In this article, a summary of MOSAiC lidar observations aboard the icebreaker Polarstern of tropospheric aerosol products is presented. Particle optical properties, i.e., light-extinction profiles and aerosol optical thickness (AOT), and estimates of cloud-relevant aerosol properties (cloud condensation nucleus, CCN, and ice-nucleating particle concentrations, INPs) are discussed, separately for the lowest part of the troposphere (near the surface at 250 m height), within the lower free troposphere (2000 m height), and regarding INPs also near the tropopause (cirrus level, 8-10 km height). In situ observations of the particle number concentration and INPs aboard Polarstern are included in the study. Strong differences between summer and winter aerosol conditions were found. During the winter months (Arctic haze period) a strong decrease of the aerosol light extinction coefficient (532 nm) with height up to about 4-5 km height was found with values of 20-100 $\mathrm{Mm}^{-1}$ close to the surface and an order of magnitude less at 1500-2000 m height. Lofted aged wildfire smoke layers caused a re-increase of the aerosol concentration from the middle troposphere up to stratospheric heights and were continuously observable from October 2019 to May 2020. In summer (June to August 2020), much lower particle extinction coefficients, frequently as low as 1-5 $\mathrm{Mm}^{-1}$, were observed. Aerosol removal, controlled by cloud scavenging processes (widely suppressed in winter, very efficient in summer) in the lowermost 1-2 km of the atmosphere, seem to be the main reason for the strong differences between winter and summer aerosol conditions. In line with this pronounced annual cycle in the aerosol optical properties, CCN concentrations



(0.2% supersaturation level) ranged from 50-500 cm$^{-3}$ in the atmospheric boundary layer (ABL) in winter and 1-40 cm$^{-3}$ in summer. In the lower free troposphere, however, the CCN level was roughly constant throughout the year with values mostly from 30-100 cm$^{-3}$. A strong contrast between winter to summer was also given in terms of ABL INPs which control ice production in low-level clouds. INP concentration of 0.01-0.2 L$^{-1}$ prevailed in the ABL in winter at typical ice-nucleating cloud temperatures of $-25$°C and assuming soil dust as the main INP type, and were roughly 2 orders of magnitude lower in

the ABL in summer at typical cloud top temperatures of $-10$°C. In the summer ABL, marine aerosol (biogenic components) is most probably the main INP type, continental INP contributions (e.g., soil dust INPs) are suppressed by efficient wet removal during long-range transport. A strong reduction in the INP population was also found in the lower free troposphere at 2000 m height from winter to summer (2 orders of magnitude), mostly due to the change in the prevailing ice-nucleation temperatures. Estimated INP concentration accumulated from 0.004-0.02 L$^{-1}$ during the winter months. The highlight of the MOSAiC lidar

studies was the detection of a persistent wildfire smoke layer in the upper troposphere and lower stratosphere from October 2019 to May 2020. The smoke particles (organic aerosol) triggered continuously cirrus formation at INP concentrations mostly from 1-20 L$^{-1}$ close to the tropopause during the entire winter period.

## 1 Introduction

The Arctic, as part of the highly polluted northern hemisphere, can no longer be regarded as a pristine environment that is

widely decoupled from the pollution centers of Asia, Europe, and North America (Abbatt et al., 2019; Willis et al., 2018, 2019; Schmale et al., 2021, 2022). The increasing number of extreme wildfires associated with long-distance transport of smoke towards all latitudes from the tropics to the North Pole is a new aspect that contributes in addition to strong changes in the environmental conditions in the Arctic (Xian et al., 2022a, b), even up to the stratosphere (Ohneiser et al., 2021; Ansmann et al., 2022). In order to consider these changes in climate modeling, especially in simulations of aerosol-cloud-precipitation inter-

actions, an improved knowledge of the aerosol conditions as a function of height and season is required. However, vertically resolved observations of aerosol properties in the Arctic are scarce, and almost absent for the winter half year.

The MOSAiC (Multidisciplinary drifting Observatory for the Study of Arctic Climate) expedition offered the unique opportunity to collect a dense data set of aerosol profiles in the North Pole region throughout a full year (Engelmann et al., 2021; Ohneiser et al., 2021). MOSAiC was the largest Arctic research initiative in history and took place from September 2019 to

October 2020. Observations were mostly performed at latitudes $> 80$°N. The goal of the MOSAiC expedition was to take the closest look ever at the Arctic as the epicenter of global warming and to gain fundamental insights that are key to better understand global climate change. A rather detailed monitoring of the atmosphere, cryosphere and biosphere in the Central Arctic was realized (see the overview articles in the MOSAiC Special Issue in Elementa) (Elementa, 2022). The German icebreaker Polarstern (Knust, 2017) served as the main MOSAiC platform for in situ observations and advanced remote sensing studies

of the atmosphere (Shupe et al., 2022). Polarstern was trapped in the ice and drifted through the Arctic Ocean from 4 October 2019 to 16 May 2020 and from 21 August to 20 September 2020 .



A state-of-the-art multiwavelength aerosol-cloud Raman lidar (Engelmann et al., 2016; Jimenez et al., 2020b) aboard Polarstern was continuously operated side by side with the ARM (Atmospheric Radiation Measurement) mobile facility 1 (AMF-1) and collected tropospheric and stratospheric aerosol and cloud profile data throughout the expedition period. Our role in the MOSAiC consortium was to provide a seasonally resolved and height-resolved characterization of aerosols and clouds in the North Pole region from the surface up to 30 km height (Engelmann et al., 2021). As one of the MOSAiC highlights, a lofted aerosol layer was continuously observed in the upper troposphere and lower stratosphere (UTLS) from about 5 to 20 km height for more than 7 months (October 2019 to mid-May 2020). The aerosol consisted of Siberian wildfire smoke in the lower part and Raikoke volcanic sulfate aerosol in the upper part of the UTLS aerosol layer (Ohneiser et al., 2021). Boone et al. (2022) provided the misleading and incorrect impression that the UTLS aerosol over the entire Arctic exclusively consisted of Raikoke sulfate aerosol. While Ohneiser et al. (2021) focused mainly on the observations of UTLS particle layer, we summarize in this article our main findings regarding tropospheric aerosols observed with the Polarstern lidar from October 2019 to September 2020.

Let us briefly outline several gaps in our knowledge about Arctic aerosols and aerosol-cloud interaction and how the MOSAiC lidar and in situ observation aboard Polarstern may contribute to this field of atmospheric research. As stated by Shupe et al. (2022), one of the main MOSAiC science questions deals with aerosol-cloud interaction: What are the processes that regulate the formation, properties, precipitation, and lifetime of Arctic clouds and what is the impact of aerosols in these processes? A proper representation of Arctic aerosols from the surface to the tropopause and a detailed consideration of complex liquid-water, mixed-phase, and cirrus cloud processes from boundary-layer to tropopause height levels in atmospheric models is of one of the key aspects to adequately simulate the water cycle in the tropospheric column and of solar and terrestrial radiation fluxes as a function of height and thus to properly simulate Arctic climate conditions and potential future changes. It is clear that surface observation (including periodic aircraft field campaigns) alone cannot provide the data needed to significantly improve our knowledge about aerosol-cloud interaction in the Arctic. Continuous, long-term vertical profiling of aerosols and cloud properties during all seasons of the year, as performed in the central Arctic during the MOSAiC expedition, are required to better meet the needs of the modeling community. Ideal monitoring conditions would be given if a network of remote sensing stations (equipped with lidars and cloud radars) (Engelmann et al., 2021; Griesche et al., 2020, 2021) could be established and combined with respective spaceborne lidar and radar observations.

Regarding Arctic aerosols, Willis et al. (2018) and Abbatt et al. (2019) provide reviews on composition and microphysical properties of Arctic aerosols as well as local and remote aerosol sources and long-range transport. These reviews are mainly based on in situ observations at ground complemented by sporadic aircraft measurements during the spring and summer months (March to September). To obtain a clear picture about the vertical layering of Arctic aerosols, their optical properties and mixing states throughout the year, Yang et al. (2021) analyzed 14 years of spaceborne CALIOP (Cloud-Aerosol Lidar with Orthogonal Polarization) observations from 2006 to 2019. In this MOSAiC study here, we continue with this effort by presenting the annual cycle of central Arctic aerosol conditions in terms of height-resolved optical and cloud-relevant properties (cloud condensation nucleus (CCN) and ice-nucleating particle (INP) concentrations). Numerous reports on the Arctic aerosol optical properties are available based on sunphotometer observations (e.g., Tomasi et al., 2012, 2015). Recently, Xian et al. (2022a, b) combined



Aerosol Robotic Network (AERONET) observations (Holben et al., 1998) with aerosol modeling to study trends and changes in the Arctic aerosol conditions during the last 20 years. However, all these photometer observations are performed from March to September, no observation from October to February are possible in the central Arctic. Lidar observations can fill this gap.

A currently very interesting research field comprises the investigations of the impact of aerosol particles on the evolution of the ice-phase in mixed-phase clouds in the lower atmosphere and in cirrus layers in the upper troposphere. Atmospheric models have especially difficulties to simulate ice-containing clouds. Schmale et al. (2022) argue that to improve the understanding of present day and future Arctic cloud processes, more detailed knowledge is needed on natural (local) Arctic aerosol emissions, their evolution and transport, and their impact on cloud microphysical properties. The situation is complicated by

long-range transport of anthropogenic aerosol pollution, agricultural and desert soil dusts, biological particles, and biogenic aerosol components from the surrounding continents. The increasing number of strong and long-lasting summer wildfires at mid and high northern latitudes, that considerably pollute Arctic regions (Xian et al., 2022a, b; Sorenson et al., 2022), further complicate modeling of Arctic climate conditions with focus on aerosol-cloud interaction.

     Significant progress has been made during the last years regarding the characterization of local Arctic ice-nucleating parti-

cles (INPs) (Creamean et al., 2018, 2019, 2022; Zeppenfeld et al., 2019; Wex et al., 2019; Hartmann et al., 2021; Li et al., 2022; Sze et al., 2022). In a remote environment such as the Arctic where particle concentrations are generally low, local production of biogenic INPs contributes significantly to the INP population in summer (Creamean et al., 2018; Wex et al., 2019) and influence ice nucleation in the lower troposphere. Creamean et al. (2019) reported a strong increase in marine INP emissions in association with phytoplankton blooms during a summertime expedition in the Bering and Chukchi Seas. Zeppenfeld et al.

(2019) observed that the increase in INP concentration in the Arctic during the summer halfyear is to a large extent related to biological activities in open waters (surface micro layer, melt ponds, open ocean polynyas). Hartmann et al. (2021) also investigated the ice nucleation properties of samples from different environmental compartments, i.e., the sea surface microlayer (SML), the bulk seawater (BSW), and fog water. In the temperature range above $-10°C$, the INP concentration was found to be the same or even higher in the Arctic than at midlatitudes latitudes, and lower than at mid latitudes at lower tempera-

tures ($< -10°C$). These recent findings point to a strong role of biogenic INPs in Arctic ice nucleating processes in summer. Alpert et al. (2022) present an INP parameterization to estimate INPs concentrations for sea spray aerosol, applicable to lidar observations.

     While soil dust particles most likely dominate ice nucleation in the lower troposphere (<2.5 km) in winter, when cloud top temperatures in mixed-phase clouds are usually far below $-15$ to $-20°C$, local INPs of biogenic origin seem to control

ice nucleation in the boundary layer in summer when cloud top temperatures in mixed-phase clouds with typical top heights below 2.5 km are usually $> -15°C$ (Griesche et al., 2021; Creamean et al., 2022). Note that ice formation requires INPs as long as temperatures are $> -38°C$. In addition to heterogeneous ice nucleation, homogeneous freezing may contribute to ice nucleation at temperatures $< -38°C$ (and thus in the upper troposphere).

     Regarding ice formation in the upper troposphere one can conclude that cirrus formation processes in polar regions are

poorly characterized by observations at all. The nucleation of first ice crystals, the formation of extended cirrus layers, and the evolution of the virga zones have a rather sensitive impact on the water cycle in the entire tropospheric column, influence the



formation of cloud layers in the middle and lower troposphere by seeder-feeder effects and thus the radiation and precipitation fields over Arctic regions in a very complex way. The limited knowledge of all these processes hinders a proper simulation of polar clouds in the climate system. The lack of knowledge is particularly acute for the winter halfyear.

The situation significantly improved in 2006 when the CALIPSO (Cloud-Aerosol Lidar and Infrared Pathfinder Satellite Observations) mission (Winker et al., 2009, 2010) and the CloudSat mission (Stephens et al., 2002) with a cloud profiling radar aboard started. Grenier et al. (2009) and Jouan et al. (2012) performed first systematic polar studies regarding the influence of aerosol particles on ice nucleation and cirrus microphysical properties based on CALIOP and CloudSat observations performed during the winter and spring seasons of 2006-2007 and 2007-2008. Jouan et al. (2012, 2014) continued with these studies
by integrating airborne in situ observations of cirrus microphysical properties in April 2008. One of the main problems is, however, that the knowledge about the influence of chemical aging and cloud processing of particles on the efficacy of these particles to serve as ice nuclei is very low (Froyd et al., 2010; Schill and Tolbert, 2013; Wolf et al., 2020). In the Arctic, the dominating aerosol types in the upper troposphere are volcanic sulfate and ash, soot particles from civil aviation, organic aerosols from wildfires, and soil dust from the surrounding continents. Long-range-transport over days and weeks may create
complex external and internal aerosol mixtures, and all these mixtures may differ significantly regarding their ice nucleating potential.

A completely new aspect, not considered in any atmospheric model yet, is the impact of wildfire smoke on cirrus formation. The role of smoke in ice-nucleation processes at temperatures from $-50°$ to $-70°C$ will be investigated in the framework of the MOSAiC data analysis. A first case study was presented in Engelmann et al. (2021). Aged wildfire smoke in the upper
troposphere and stratosphere consists to more than 95% of organic material (Yu et al., 2019; Kablick et al., 2020; Khaykin et al., 2020; Torres et al., 2020; Ohneiser et al., 2023). Only 2-3% is black carbon. Thus, the organic material determines the ice-nucleating properties of smoke particles, and not the soot fraction. Jahn et al. (2020) and Jahl et al. (2021) hypothesized that aged smoke particles contain minerals and that these components determine the smoke INP efficacy. How relevant this aspect is remains to be shown. Such smoke particles should be able to trigger ice nucleation in mixed-phase clouds at comparably
high temperatures of $-20$ to $-30°C$ already.

In this article, we will summarize our main MOSAiC results regarding the optical and cloud-relevant properties of tropospheric aerosols and will contribute in this way to several modern Arctic research aspects just mentioned. The article is organized as follows. In Sect. 2, the applied instrumentation and data analysis methods are described. Several case studies of tropospheric aerosol profiling are discussed in Sect. 3.1 documenting long-range aerosol transport towards high northern lati-
tudes during the winter and summer months. Case studies for the winter half year during the Arctic haze period were presented in detail already by Engelmann et al. (2021). The annual cycle of tropospheric aerosol profiles and 532 nm AOT observed during MOSAiC year 2019-2020 are then discussed in Sect. 3. Time series of in-situ-measured and lidar-derived particle number concentration, used as proxy for CCN, and of INP concentration for different height levels in the troposphere are given in Sect. 4. A short summary and concluding remarks complete the study in Sect. 5.



## 2 MOSAiC instruments and data analysis

### 2.1 MOSAiC Polarstern route

The full track of the Polarstern is given in Creamean et al. (2022), Shupe et al. (2022), and Boyer et al. (2023). The ice breaker drifted with the ice through the central Arctic at latitudes $\geq 85°$N until the beginning of April and cruised betwee 83-84°N until 22 May 2020. Because of the COVID-19 pandemic Polarstern had to leave the ice zone and to transit to Ny-Ålesund (78.9°N, 11.9°E) on the island of Spitsbergen in Svalbard, Norway, in the beginning of June 2020 to exchange science team members. The same procedure was necessary in the beginning of August 2020. As a consequence of theses complications, from June to mid of August 2020, the observations were restricted to latitudes around 80-82°N. From mid August to the end of September 2020, observations were again taken at latitudes $\geq 85°$N.

### 2.2 Lidar instrument: Polarstern Polly

The remote sensing infrastructure aboard Polarstern was discussed in Engelmann et al. (2021). The multiwavelength polarization Raman lidar Polly (POrtabLe Lidar sYstem) (Engelmann et al., 2016) performed measurements from 26 September 2019 to 2 October 2020 (Polly, 2022). A detailed description of the Polly instrument with all the upgrades realized during the last years can be found in Hofer et al. (2017) and Jimenez et al. (2020b). The Polly instrument is mounted inside the OCEANET-Atmosphere container of the Leibniz Institute for Tropospheric Research (TROPOS). This container is designed for routine operation aboard Polarstern between Bremerhaven, Germany, and Cape Town, South Africa, and Punta Arenas, Chile (Kanitz et al., 2011, 2013; Bohlmann et al., 2018; Yin et al., 2019), and was operated for the first time in the Arctic during a two-month campaign in June and July 2017 (Griesche et al., 2020, 2021).

### 2.3 MICROTOPS II sunphotometer

A handheld MICROTOPS II sunphotometer (Ichoku et al., 2002) was used by the TROPOS lidar team aboard Polarstern to measure the AOT at 440, 500, 870, and 1020 nm wavelength from June to September 2020 when ever possible to support lidar observations of particle extinction profiles. MICROTOPS II is the standard device of MAN (Maritime Aerosol Network) (Smirnov et al., 2009) which is a component of AERONET (Holben et al., 1998). An operator is required to point the photometer to the Sun for a while to take stable measurements. Continuous, unattended measurements are not possible. The data are stored in the MAN (Maritime Aerosol Network) data base (AERONET-MAN, 2022).

### 2.4 Arctic CALIOP observations

The Cloud-Aerosol Lidar with Orthogonal Polarization (CALIOP) is a spaceborne polarization lidar that performs global profiling of aerosols and clouds (Winker et al., 2009, 2010). The satellite lidar was launched in April 2006. In this study, we will compare our Arctic aersol observations (optical properties) with results published by Yang et al. (2021). These authors analyzed all Arctic CALIOP aerosol data for latitudes >65°N from 2006 to 2019. The maximum latidude covered by the



CALIOP observations is 81.8°N. Yang et al. (2021) present Arctic maps of 532 nm AOT, time series of monthly resolved 15-year mean AOT and seasonally resolved 15-year mean height profiles of the particle extinction coefficient for dominating aerosol types.

## 2.5 Lidar-derived particle optical properties

An overview of all measured and retrieved lidar products is given in Table 1 in Engelmann et al. (2021). The lidar products
together with typical uncertainties used in this study are listed in Table 1 of this paper. The basic aerosol observations comprise height profiles of the particle backscatter coefficient at 355, 532, and 1064 nm, the particle extinction coefficient at 355 and 532 nm, the respective extinction-to-backscatter ratio (lidar ratio) at 355 and 532 nm, and the particle linear polarization ratio at 355 and 532 nm (Baars et al., 2016; Hofer et al., 2017; Ohneiser et al., 2021). Lidar signals are measured with a near-range and a far-range telescope, covering different height ranges so that backscatter coefficients and depolarization ratios are measurable
from about 100 m to 30 km, and extinction coefficients and lidar ratios from about 400 m upward. The main features of the basic MOSAiC aerosol data analysis (including signal correction, Rayleigh backscattering and extinction correction, temporal averaging and vertical smoothing of signal profiles) are described in Ohneiser et al. (2020, 2021, 2022).

## 2.6 POLIPHON Arctic aerosol model: optical vs microphysical properties

The POLIPHON (Polarization Lidar Photometer Networking) method is a robust and practicable lidar method to derive number,
surface area, and volume concentrations of particles from the measured optical properties in the troposphere and stratosphere and to estimate tropospheric CCN and INP concentrations (Mamouri and Ansmann, 2016, 2017; Ansmann et al., 2019a, 2021). The POLIPHON method makes use of the height profiles of the 532 nm particle backscatter coefficient and particle depolarization ratio and converts the measured optical into microphysical properties by using specific aerosol-type-dependent particle models. These models are derived from long-term AERONET (Aerosol Robotic Network) observations (Holben et al., 1998)
around the globe and connect the optical and underlying microphysical properties for well-defined aerosol types, such as soil dust, marine particles, anthropogenic haze, and wildfire smoke.

In the framework of the MOSAiC data analysis, Arctic AERONET observations were used to develop a respective aerosol model for Arctic aerosol particles, i.e., a mixture of aged urban haze, biomass burning smoke, and soil dust after long-distance transport and a minor contribution of marine particles. Sun and sky photometer observation of 11 Arctic AERONET stations
covering up to almost 25 years of observations (1997-2021) were considered in this project (AERONET, 2022). According to the AERONET observations, the Arctic aerosol shows remarkably constant properties from the spring season to the late summer season. Typical Ångström exponents (for the 440-870 nm spectral range) are 1.4-1.6, clearly indicating non-marine aerosol components. The fine-mode fraction is around 0.9 and indicates the dominance of anthropogenic pollution and biomass-burning smoke. Most of the time the aerosol optical thickness (AOT) is found in the range of 0.015-0.15 at 500 nm which is in
good agreement with the studies of Tomasi et al. (2012, 2015) and Xian et al. (2022a).

To obtain height profiles of Arctic aerosols in terms of standard products such as volume concentration $v(z)$, surface area concentration $s(z)$, and particle number concentrations $n_{\mathrm{rmin}}(z)$ considering all particles with radius > rmin (in nanometer),



the following basic relationships are available:

$$v(z) = c_v L \beta(z), \tag{1}$$

$$s(z) = c_s L \beta(z), \tag{2}$$

$$n_{rmin}(z) = c_{rmin} L \beta(z), \tag{3}$$

with the particle backscatter coefficient $\beta(z)$ at height $z$ and the extinction-to-backscatter or lidar ratio $L$. Typical lidar ratios for Arctic aerosol are 55±15 sr at 532 nm. These lidar ratios indicate continental fine-mode aerosol (Mattis et al., 2004) and are in agreement with the high Arctic Ångström exponents of 1.4-1.6. Lidar ratios for a clean marine environment are around 20-25 sr at 532 nm (Groß et al., 2015, 2016) and >70 sr for strongly light-absorbing wildfire smoke particles (Haarig et al., 2018; Ohneiser et al., 2020, 2022). The extinction-to-volume conversion factor $c_v$, the extinction-to-surface-area conversion factor $c_s$, and the extinction-to-number conversion factors $c_{rmin}$ for 532 nm are obtained from the analysis of the Arctic AERONET observations regarding the relationship between measured aerosol optical and retrieved microphyscial properties following the procedure described by Mamouri and Ansmann (2016, 2017). In contrast to the approach of Mamouri and Ansmann (2016), we applied a simple linear regression analysis to the extinction-vs-number-concentration data field to obtain, e.g., $c_{65}$ and $c_{85}$ in Table 2, instead of a regression (of particle extinction vs $n_{65}$ and $n_{85}$) in logarithmic scales. Table 2 shows several conversion factors for Arctic aerosols. These conversion factors are required to estimate particle number concentrations and surface area concentrations, which are used as input in the estimation of CCN and INP concentrations as explained in Sects. 2.7 and 2.8.

Input in these CCN and INP retrieval procedures are aerosol parameters for dry conditions, i.e., aerosol properties that can be observed at ambient conditions only if the relative humdity (RH) is clearly below 40%. However, AERONET sunphotometer observations in the Arctic are typically performed at RH around 80% in the lower, aerosol-laden atmosphere according to the MOSAiC 2019-2020 radiosonde observations (Maturilli et al., 2021), in good agreement with the study of Shupe et al. (2011) at Arctic land-based observatories. So, all the conversion factors are derived for aerosol scenarios observed at high humidity of RH=80% and should therefore be applied to lidar-derived extinction coefficients (and corresponding particle size distribution scenarios) at RH=80%.

The aerosol particles thus contain a considerable amount of water at high humidity so that the aerosol backscatter and extinction coefficients are significantly enhanced compared to respective optical properties for dry conditions. To obtain the dry aerosol parameters (e.g., $n_{50,dry}$ needed in the CCN estimation, $s_{dry}$ as input in the INP retrieval) the following procedure was necessary to correct for water uptake effects: We make use of the well-known so-called enhancement factor $(1 - RH/100\%)^\gamma$ with RH in % and an exponent $\gamma$ of −0.46 for continental fine-mode particles (Skupin et al., 2016). The enhancement factor relates the optical properties of the particles measured at ambient RH conditions (e.g., at 80%) to respective values for dry conditions (e.g., RH of 0-20%). In the first step, we converted the lidar profiles of the particle extinction coefficient for ambient RH (known from the MOSAiC radiosonde RH profiles) to values for RH=80% by multiplying the measured extinction values with the factor $(1 - 80/100\%)^{-0.46}/(1 - RH/100\%)^{-0.46}$. Then, we multiplied these extinction coefficients for RH=80% with the conversion factor of $c_{85}$ to obtain an estimate for the height profile of the particle number concentration $n_{85}(z)$ at RH=80%. This number concentration $n_{85}$ was then interpreted as an appropriate proxy of $n_{50,dry}$ after removal of the water uptake effect.



It is assumed in this way, that water uptake causes an increase of the radius of dry particles by roughly a factor of 1.5 when RH is increased from low RH to high RH values around 80%.

In order to obtain the height profile of the particle surface area concentration $s_{\mathrm{dry}}(z)$ for Arctic aerosols, we used the
computed lidar extinction profiles for RH=80% and multiplied these profiles with the conversion factor $c_{\mathrm{s}}$ to obtain the surface-area profile $s(z)$ for RH=80%. Then we converted this $s$ profile to a profile for RH=20% by multiplying all $s$ values with the factor $(1-20/100\%)^{-0.46}/(1-80/100\%)^{-0.46}$. This profile, after water uptake correction, was interpreted as $s_{\mathrm{dry}}$.

In the case of the CCN and INP retrieval in the Arctic atmospheric boundary layer (ABL) in summer, we assume that local marine aerosols prevail in the rather shallow ABL and use the conversion factors for marine aerosol (Mamouri and Ansmann,
2016), e.g., a marine conversion factor $c_{\mathrm{s,m}} = 2.52\,\mathrm{Mm}\,\mu\mathrm{m}^2\,\mathrm{cm}^{-3}$ (for ambient marine conditions which are also characterized by typical RH values of 80%). The same procedure as described above was applied to remove the water uptake effect to end up with $n_{50,\mathrm{dry}}$ and $s_{\mathrm{dry}}$. However, here we used $\gamma = -0.82$ for pure marine aerosol (Haarig et al., 2017) and further assumed that $n_{100}(z)$ for RH=80% can be interpreted as an appropriate proxy of the marine $n_{50,\mathrm{dry}}$ (Mamouri and Ansmann, 2016).

According to Table 1, the microphysical aerosol properties (dry volume and surface-area concentrations) can be estimated
with an uncertainty of 25%. The uncertainty is of the order of 50% in the case of the $n_{50,\mathrm{dry}}$ retrieval when the aerosol type is well known as comparisons with airborne in situ measurements of CCN concentrations showed (Düsing et al., 2018; Choudhury et al., 2022). The uncertainty is larger (within a factor of 2) when the aerosol type (including the aerosol size distribution for this aerosol type) is not well known or rather complex mixtures of different hygroscopic and hydrophobic, fine and coarse aerosol particles prevail (Haarig et al., 2019; Georgoulias et al., 2020).

In Sect. 4.3, we present INP time series for wildfire smoke particles at cirrus level. In this approach, we use the conversion factors for aged wildfire smoke, obtained from AERONET observations at dry UTLS conditions (Ansmann et al., 2021). In this approach, we converted the optical properties measured at upper tropospheric humidity conditions to values for RH=20% first, and then estimated the surface area concentration by using the smoke-related conversion factors (Ansmann et al., 2021). This smoke-related dry-particle surface area concentration was then used as input in the smoke INP parameterization explained in
Sect. 2.8.

## 2.7 CCN estimation

In Sect. 4.1, lidar-derived time series of the CCN concentration $n_{\mathrm{CCN}}$ at 250 and 2000 m height are presented. CCN values at 250 m height may well represent the aerosol conditions during low level cloud formation at the top of the ABL. According to Peng et al. (2023), ABL top height was mostly around 200 m over the Polarstern during the MOSAiC year. Time series
at 2000 m height provide insight into the CCN conditions in the lower free troposphere where stratiform mixed-phase cloud layers frequently develop.

As discussed in Mamouri and Ansmann (2016), the particle number concentration $n_{50,\mathrm{dry}}$ can be used as proxy for $n_{\mathrm{CCN}}$ in an air parcel in which the relative humidity over water is 100.2% (supersaturation level of 0.2%, $S_{\mathrm{WAT}} = 1.002$) so that



droplets form:

$$n_{\mathrm{CCN}}(S_{\mathrm{WAT}}) = f_{\mathrm{ss}} \times n_{50,\mathrm{dry}}. \tag{4}$$

The factor $f_{\mathrm{ss}}$ is set to 1.0 for a water supersaturation value of 0.2% and is introduced to estimate CCN concentrations for lower and higher supersaturation levels. Values for $f_{\mathrm{ss}}$ were found to be about 0.4, 1.5, and 2.0 for $S_{\mathrm{WAT}} = 1.001$, 1.004, and 1.007, respectively, in the Canadian Arctic (Tuktoyaktuk, 69.4°N, 133.0°W) in the spring of 2014 (Herenz et al., 2018). According to their observations the critical diameter $d_{\mathrm{crit}}$ was 107 nm at $S_{\mathrm{WAT}} = 1.002$. For the critical diameter $d_{\mathrm{crit}}$, the integral over the independently measured particle size distribution from $d_{\mathrm{crit}}$ to the maximum size bin, $d_{\mathrm{max}}$, is equal to the measured CCN concentration $n_{\mathrm{CCN}}$. $d_{\mathrm{crit}}$ decreases with increasing supersaturation. Also Dada et al. (2022) derived a critical diameter around 100 nm for a supersaturation of 0.2% from MOSAiC observation aboard Polarstern during a warm airmass intrusion event in April 2020. All these findings corroborate that $n_{50,\mathrm{dry}}$ is an appropriate proxy for $n_{\mathrm{CCN}}$ for the supersaturation level of 0.2%.

## 2.8 INP estimation

In Sect. 4.2 and in Sect. 4.3, we present MOSAiC time series of lidar-derived INP estimates for the height levels of 250 m, 2000 m, and 1 km below the tropopause. INP time series for 250 and 2000 m show the ice-nucleation conditions of mixed-phase clouds in the ABL and lower free troposphere, and the INP concentration values for the uppermost troposphere indicate the potential of aerosol particles to influence ice nucleation at cirrus level.

The most important ice-nucleating aerosol types in the lower troposphere (heterogeneous ice nucleation regime) of the Arctic, with cloud top temperatures $> -35$°C, seem to be soil dust particles in winter as well as in summer (in the free troposphere), and sea spray aerosol (SSA) particles, carrying ice active substances of biogenic origin, in the summer ABL. Observations showed that immersion freezing is the dominating ice nucleation process in mixed-phase cloud formation at temperatures $\geq -30$°C (Ansmann et al., 2008, 2009; de Boer et al., 2011; Westbrook and Illingworth, 2011, 2013). In the case of immersion freezing, ice nucleation is initiated on particles immersed in the liquid-water droplets. In the following Sects. 2.8.1 and 2.8.2, we briefly describe the INP retrievals for these two INP types. Furthermore, the INP parameterization for wildfire smoke at cirrus level is given in Sect. 2.8.3.

### 2.8.1 Arctic aerosol of continental origin

It is assumed that the Arctic aerosol contains a few percent of agricultural and desert soil dust particles which solely serve as INPs. Zhao et al. (2022) recently discussed the long-range transport of desert dust from Asia to the Arctic and show that dust must be expected everywhere over the Arctic in the tropospheric column from the surface up to the tropopause. The studies of Yang et al. (2021) and Xian et al. (2022a) support this assumption. To estimate the dust-related INP concentration $n_{\mathrm{INP}}$ we applied the INP parameterization of Ullrich et al. (2017). Besides the temperature, at which the ice nucleation is initiated, the aerosol surface area $s_{\mathrm{dry}}$ and the dust fraction (i.e., the fractional contribution to the particle surface area concentration) are required as input.





315    The Ullrich et al. (2017) parameterization was carefully compared with directly observed INP concentrations during pure Saharan dust conditions over Cyprus in 2016 (Marinou et al., 2019). Compared to the in situ INP observations, the immersion freezing parameterization of Ullrich et al. (2017) overestimated $n_{INP}$ by a factor of 10-50 at temperatures from $-15$°C to $-25$°C (Marinou et al., 2019). A similar observation regarding overestimation was made by Knopf et al. (2021) and Wieder et al. (2022). This overestimation is considered in the INP values shown in Sect. 4.2.

320    We ignore aging and coating of dust particles with sulfate or organic material which may reduce or enhance the INP efficiency. As pointed out in the review article of Willis et al. (2018), aerosol particles can undergo significant chemical aging and cloud processing along the transport path to Arctic regions. Aged dust particles may be partly or even completely coated with sulfate or organic substances. The potential to serve as INP may then be significantly reduced by 1-2 orders of magnitude for a given scenario of meteorological and aerosol conditions depending on the thickness of the coating (Möhler et al., 2008; Cziczo et al., 2009; Kulkarni et al., 2014; Kanji et al., 2017, 2019; Knopf et al., 2018). Even a complete suppression of ice nucleation can not be excluded in the case of thick coatings.

As indicated in Table 1, we assume that the lidar observations allow us at least to obtain the order of magnitude of occurring INP concentrations. The large uncertainty is caused by the applied INP parameterization and not by the aerosol input parameters obtained from the lidar observations. To check the reliability of the INP retrieval procedures we therefore make use of so-called closure studies in which the lidar-derived INP concentration $n_{INP}$ is compared with estimated ice crystal number (ICN) concentration $n_{ICE}$ from lidar-radar observations (Ansmann et al., 2019b; Marinou et al., 2019; Engelmann et al., 2021). Good agreement in these closure studies, i.e., similar estimates of $n_{INP}$ and $n_{INP}$, in the absence of secondary ice production (Ramelli et al., 2021), then indicates a high reliability of the selected INP parameterization. This closure approach will be applied also within the planned MOSAiC studies (to be presented in follow-up articles).

### 2.8.2  Sea spray aerosol (SSA) in the summer ABL

Recently, Alpert et al. (2022) introduced a parameterization that permits the estimation of SSA INP concentrations caused by biogenic aerosol components. These INPs may be responsible for the observed ice nucleation in the summer ABL at high temperatures $> -10$ to $-15$°C (Griesche et al., 2021), at which mineral dust particles are no longer efficient INPs. The water-activity-based immersion freezing model ABIFM (Knopf and Alpert, 2013), drawn from the water-activity-based homogeneous ice nucleation theory (Koop et al., 2000) is used here to predict INP concentrations. The calculation procedures to estimate INP concentrations (following the ABIFM approach), with special focus on lidar observations, are compiled in Ansmann et al. (2021) and applied to a MOSAiC smoke-cirrus case in Engelmann et al. (2021).

In the first step, the ice-nucleation rate $J_{het}$ is computed as a function of SSA specific parameters k = 26.6132 and b = -3.9346 (Alpert et al., 2022) and for a given ice-nucleation temperature and ice supersaturation $S_{ICE}$. The product of $s_{dry} \times J_{het} \times \Delta t$ then yields $n_{INP}$. Ice nucleation takes place during the time interval $\Delta t$. In the case of mixed-phase clouds in the Arctic ABL we can assume a relative humidity of 100% within the cloud layer so that ice supersaturation is fixed for a given ice-nucleation temperature. Essential variable input parameters in the INP computations are thus the SSA surface area concentration $s_{dry}$, the





ice-nucleation temperature and the duration $\Delta t$ during which ice nucleation occurs. A realistic duration of an ice nucleation event is $\Delta t$ of 10 minutes.

### 2.8.3   Aged wildfire smoke and soil dust in the upper troposphere

In the upper tropopshere, wildfire smoke particles dominated over the High Arctic from October 2019 to May 2020 (Ohneiser et al., 2021) and also during late summer (September 2020). The ice nucleation efficiency of aged smoke particles is determined by organic material (organic carbon, OC). The black carbon (BC) or soot content is typically 2-3% (Dahlkötter et al., 2014; Yu et al., 2019; Torres et al., 2020; Ohneiser et al., 2023) and has no relevant impact on the ice-nucleating efficiency of aged wildfire smoke particles. Biomass-burning particles also contain humic like substances (HULIS) which represent large macromolecules that could serve as INP at low temperatures of $-50$ to $-70°C$ (Wang and Knopf, 2011; Wang et al., 2012; Knopf et al., 2018).

Because of the complex chemical, microphysical, and morphological properties of aged fire smoke particles, which can occur as glassy, semi-liquid, and liquid aerosol particles, the development of smoke INP parameterization schemes is a crucial task (Knopf et al., 2018). The particles and released vapors in young biomass burning plumes undergo chemical and physical aging processes on their way up to the tropopause and during long-range transport over weeks and months. Aging includes photo-chemical processes, heterogeneous chemical reactions on and in the particles, condensation of gases on the particle surfaces, collision and coagulation, and cloud processing (when acting as CCN or INPs in several consecutive cloud evolution and dissipation events). All these impacts change the chemical composition of the smoke particles, their morphological characteristics (size, shape, and internal structure), and the internal mixing state of the smoke particles.

It is assumed that smoke particles, after finalizing the aging process, show an almost perfect spherical core-shell structure with a BC-containing core and an OC-rich shell, and that the ability to serve as INP mainly depends on the material in the shell and thus on the organic material of the particles. If the particles are in a glassy state, they can serve as deposition ice nucleation (DIN) INPs (Murray et al., 2010; Wang and Knopf, 2011; Wang et al., 2012). DIN is defined as ice formation occurring on the INP surface by water vapor deposition from the supersaturated gas phase. When the supercooled smoke particles can take up water or their shell deliquesces, immersion freezing can proceed, where the INPs immersed in an aqueous solution can initiate freezing (Wang et al., 2012; Knopf and Alpert, 2013; Knopf et al., 2018). If the smoke particle become completely liquid (and no insoluble material within the particles is left), homogeneous freezing will take place at temperatures below  235 K (Koop et al., 2000). In the case of cirrus formation during the MOSAiC winter months, we used the ABIFM (immersion freezing model) to compute $n_{INP}$ (Engelmann et al., 2021).

Input parameters in the INP computations are the surface area concentration $s_{dry}$ for dry smoke particles, the ice-nucleation temperature, ice supersaturation, the duration $\Delta t$ of, e.g., an updraft event of a gravity wave, producing the assumed ice supersaturation, and finally the organic material in the liquid shell of the smoke particles. $\Delta t$ was set to 600 s, a typical temporal length of the lifting period of a gravity wave (Kalesse and Kollias, 2013). Regarding the organic material, we chose to apply the ABIFM for leonardite (a standard humic acid surrogate material) to represent the amorphous organic coating of smoke particles. Leonardite, an oxidation product of lignite, is a humic-acid-containing soft waxy particle (mineraloid), black





or brown in color, and soluble in alkaline solutions. The INP characteristics of leonardite were studied in detail in laboratory experiments (Knopf and Alpert, 2013; Rigg et al., 2013). It served as surrogate for humic-like substances in extended immersion freezing laboratory studies.

During the summer months (June-August 2020, after the dissolution of the wildfire smoke layer), we assumed that particle ensembles found in the upper tropopshere were a mixture of anthropogenic aerosol, biomass burning smoke, and agricultural and desert soil dust. In the presence of dust, these particles will dominate heterogeneous ice nucleation (at -50° to-70°C) because they are ice-active at much lower ice supersaturations than, e.g., smoke particles (as will be discussed in Sect. 4.3). To estimate the ice-nucleating potential of dust particles at the tropopause level, we applied the DIN INP parameterization of

Ullrich et al. (2017). Input parameter in this estimation is the particle surface area concentration $s_{\mathrm{dry}}$, the fractional contribution of dust to the surface area concentration, air temperature, and ice supersaturation $S_{\mathrm{ICE}}$. The INP prediction by means of the DIN parameterization of Ullrich et al. (2017) was found to be in excellent agreement with the in situ dust INP observations in pure dust over Cyprus (Marinou et al., 2019). Cirrus closure experiments discussed in Ansmann et al. (2019b) corroborated the findings of (Marinou et al., 2019).

**2.9  Instrumentation for in situ measurements of aerosol microphysical properties and INP concentrations aboard Polarstern**

Continuous in situ observations of the dry particle number concentration, particle number size distribution and BC mass concentration (Boyer et al., 2023) as well as of INP concentrations (Creamean et al., 2022) were performed aboard Polarstern during the entire MOSAiC period from October 2019 to September 2020. Results will be shown in Sects. 4.1 and 4.2 together

with respective lidar estimates. A detailed description of the applied instruments and data analysis methods is given in the articles of Boyer et al. (2023) and Creamean et al. (2022).

The number concentrations $n_{50,\mathrm{dry}}$ measured in situ (Boyer et al., 2023) consider particles with diameters from 100 to 500 nm and is also used as a proxy for the CCN concentration $n_{\mathrm{CCN}}$ for $S_{\mathrm{WAT}} = 1.002$. CCN concentrations were also measured aboard Polarstern (Dada et al., 2022) and will be included in the discussions of the MOSAiC observations in Sect. 4.1.

The in situ observations of $n_{50,\mathrm{dry}}$ were carefully checked and corrected for contamination by local pollution (exhaust plume of Polarstern and further aerosol sources on the near-by measurement field station on the pack ice) (Beck et al., 2022). About 40% of the measured data had to be removed (Boyer et al., 2023).

In contrast to the lidar-derived INP estimates (assuming only dust particles as INPs), the in-situ-measured INP concentrations cover all (dust and non-dust) particle types that can contribute to ice nucleation. Particle sizes (diameter) from 150 nm to 6 $\mu$m

are considered in the INP observations.

**2.10  Air mass source analysis**

Ensemble backward trajectories are computed (as part of case studies) by using the NOAA (National Oceanic and Atmospheric Administration) HYSPLIT (HYbrid Single-Particle Lagrangian Integrated Trajectory) model (HYSPLIT, 2022; Stein et al., 2015; Rolph et al., 2017). The arrival heights are set into observed aerosol layers to identify the origin of the pollution.





Furthermore, the aerosol-source attribution method of Radenz et al. (2021) was applied. This air mass identification tool was developed to support the interpretation and evaluation of lidar profiles. The normalized (accumulated) residence time, during which the air masses traveled within the well-mixed boundary layer at heights below 2 km, is computed before the air masses cross Polarstern at well specified arrival heights (from the surface to 12 km with a resolution of 500 m). This analysis is also based on HYSPLIT backward trajectories. 10 d backward trajectory analysis was found to be sufficient to identify

the continental pollution sources (Asia, Europe, or North America), or, in cases with background aerosol conditions that the respective air masses obviously did not cross any populated continental region (aerosol source region) during a period longer than a week before arrival over Polarstern.

## 3 Aerosol layering and aerosol optical properties

### 3.1 Clean and polluted conditions during the MOSAiC summer: case studies

Before we discuss the annual cycle of the aerosol conditions during the MOSAiC year, several contrasting cases with clean and polluted conditions measured during summer 2020 are presented. Arctic haze events observed in February and March 2020 were already discussed in Engelmann et al. (2021). Figure 1 shows height profiles of basic lidar products, i.e., of the particle backscatter and estimated extinction coefficient at 532 nm, for three days during the summer up to the fall freeze-up period. On all three days the lowest part of the troposphere was rather clean. On 30 June 2020, Arctic background conditions

were observed over the Polarstern with extinction coefficients of 1-3 $Mm^{-1}$. The backscatter peak at the surface was probably caused by weak fog which drifted over the lidar during the signal averaging period (18-24 UTC). The lidar-derived 532 nm AOT was 0.023 on 30 June 2020 (when ignoring the fog-related near-surface backscatter peak). The MICROTOPS photometer measured a 500 nm AOT of 0.035 in the evening of 30 June 2020. According to the HYSPLIT backward trajectory analysis in Fig 2a the airmass was not in contact to any populated region during the last 10 days. Such clean conditions in the lowest

5-7 km height were frequently observed from the end of May to mid-July 2020.

On 5 August 2020, the atmosphere was significantly polluted above 1.5 km height. HYSPLIT backward trajectories in Fig 2b indicate air mass transport from central and eastern Siberia at 2 km height. The same holds for 4 km height. The source identification method developed by Radenz et al. (2021) was applied in Fig. 3 to identify the aerosol sources for all heights in the troposphere. The length of each bar for the different heights indicates the time that the air mass spent at heights below 2 km

during the long-distance travel and thus were able to accumulate aerosol pollution over the Arctic Ocean, adjacent continental sites (savanna and shrubland at high latitudes), and regions further south (grass/cropland). As can be seen, the impact of continental airmasses increased with height and time. The air masses above 1.0 km (arriving at 18 and 21 UTC) were able to significantly uptake anthropogenic pollution, smoke and dust particles over Siberia. The MICROTOPS 500 nm AOT was close to 0.05 on 5 August. The integration of the lidar extinction profile yields a 532 nm AOT of 0.047. By combining AOT (from

MICROTOPS) and column backscatter (BC from lidar) we obtain a column lidar ratio (AOT/CB) of 56.6 sr, a typical value for anthropogenic haze (Mattis et al., 2004). The Ångström exponent (MICROTOPS AOT, 440-870 nm) was around 1.7-1.9 in the evening of 5 August and thus in good agreement with the backscatter-related Ångström exponent (355-1064 nm) of 1.4-2 in





the height range from 2-6 km as shown in Fig. 1b. The particle depolarization ratio was low (0.02-0.03) which is indicative for an almost dust-free air mass.

On 10 September 2020, a pronounced haze layer between 1.2 to 3.5 km was observed. HYSPLIT backward trajectories for this case are shown in Fig. 2c and indicate a pollution transport mainly from northern and western Europe and North America. Polarstern was close to 89°N on this day. The AOT of the pronounced haze layer was 0.03, the overall AOT close to 0.035. By combing MICROTOPS AOT and lidar-derived column backscatter we obtained a column lidar ratio of 57.8 sr, again a characteristic value for anthropogenic pollution. The moderately low Ångström exponent (MICROTOPS AOT, 440-870 nm)
of 1.3 and around 1.4 (lidar, backscatter, 355-1064 nm) together with the enhanced particle depolarization ratio of 0.05-0.07 indicate a noticable contribution of coarse-mode dust of about 5% to the backscatter and extinction coefficients.

It is notworthy to mention that the Arctic haze layers in winter showed the highest aerosol burden in the lowest 500-1000 m of the troposphere with highest extinction coefficient of the order of 30-70 $Mm^{-1}$ close to the surface, as will be discussed in the next section. The contribution of the lowest 1 km to the total tropospheric 532 nm AOT was typically 0.03-0.05 in winter.
In summer, these near-surface haze layers are absent, probably as a result of very efficient wet removal by low-level clouds, drizzle, fog, and liquid-water precipitation (Browse et al., 2012). The remaining AOT for the lowest 1000 m of the atmosphere is of the order of 0.002-0.004 in Fig. 1a and thus an order of magnitude lower than a typical marine AOT over the open ocean at midlatitudes.

Fig. 4 shows a pyroCb-lofted wildfire smoke layer in the upper troposphere measured on 19 September 2020. Associated
HYSPLIT backward trajectories are shown in Fig. 5. Fires in northern Canada or Alaska were responsible for the smoke pollution. The enhanced volume depolarization ratio of about 5% in Fig. 4a and the corresponding particle depolarization ratios of 6-7% (not shown) indicate irregularly shaped smoke particles. The depolarization feature is typical for fast smoke lofting by pyroCb convection (Haarig et al., 2018; Ohneiser et al., 2020). PyroCb convection can transport smoke plumes within less than one hour into the upper troposphere (Rodriguez et al., 2020; Reisner et al.), which is then the aerosol source
region in the HYSPLIT trajectory analysis. Because of the fast lofting there is not sufficient time for aging and the development of an ideal, spherical core-shell structure which would cause rather low depolarization ratios. The smoke layer showed high extinction coefficients close to 300 $Mm^{-1}$ and produced a large 532 nm AOT of 0.4.

### 3.2 MOSAiC annual cycle: Profiles of backscatter and extinction coefficients

The annual cycle of aerosol optical properties during the MOSAiC year is shown in Figs. 6 and 7. Monthly and 2-month mean
backscatter and extinction profiles are given. Figure 6 provides an overview of the year-around observations up to 20 km height. One of the MOSAiC highlights was the detection of a pronounced and persistent wildfire smoke layer in the upper troposphere and lower stratosphere (UTLS) over the North Pole region from October 2019 to May 2020. This unique event was discussed in detail by Ohneiser et al. (2021). The smoke originated from record-breaking wildfires in central and eastern Siberia, north of Lake Baikal, in the summer of 2019 (Johnson et al., 2021; Ohneiser et al., 2021, 2023; Xian et al., 2022b; Sorenson et al., 2022)
and affected the ozone layer (Ohneiser et al., 2021; Voosen, 2021; Ansmann et al., 2022). The smoke reached the UTLS height range most probably by self-lofting processes (Ohneiser et al., 2021, 2023). Large amounts of Siberian smoke were transported





into the central Arctic at all heights from mid-June to mid-August 2019 and caused extremely large 500-550 nm AOT values exceeding daily mean values of 0.2 in the polar region from 70° to 90°N from 24 July to 22 August 2019 (Xian et al., 2022b). Such high AOTs as observed in August 2019 were never observed before over the polar region, the authors stated. However, at
the end of September 2019, when the MOSAiC journey of Polarstern started in northern Norway, the lower troposphere over the central Arctic was widely cleaned from this pollution. Only the UTLS smoke layer remained.

Figure 7 shows the same MOSAiC profiles as in Fig. 6, however, now up to 10 km height in terms of the particle extinction coefficient. The 10 km height level is close to the tropopause level. The backscatter coefficients in Fig. 6 were multiplied by a typical extinction-to-backscatter ratio (lidar ratio) of 55 sr. This lidar ratio holds well for mixtures of continental haze, mineral
dust, and wildfire smoke as was shown in the foregoing Sect. 3.1 (5 August and 10 September 2020 case studies). The lidar ratio may vary between 40-70 sr, thus the uncertainty in the extinction values is of the order of 20%. In the summer (June-July, August-September profiles in Figure 7), the shown extinction values may, however, be too large (by a factor of 2) at heights <1 km. As was shown in Fig. 1a, clean marine conditions prevailed during the lidar observations in the lowest part of the atmosphere. The lidar ratio for marine aerosol is 20-25 sr (Groß et al., 2015, 2016; Haarig et al., 2017) and thus roughly a
factor of 2 lower than the one for continental aerosols. Therefore, the summer extinction values in Fig. 7 for heights up to 1 km were probably a factor of 2 too high.

The most striking feature in Fig. 7 is the strong decrease of the particle extinction coefficient with height during the winter months (Arctic haze season) when aged anthropogenic aerosol, soil dust, and biomass burning smoke is transported into the Arctic from the surrounding continents (North America, Asia, Europe) (Stohl, 2006; Willis et al., 2018; Engelmann et al.,
2021). Most of the pollution reaching Polarstern at lower heights in winter 2019-2020 originated from northern Asia (Creamean et al., 2022; Boyer et al., 2023). Arctic haze events observed on 4 February and 4 March 2020 were discussed in Engelmann et al. (2021). The largest extinction coefficients occurred close to the surface where the extinction values (measured with lidar at ambient humidity conditions) were as high as $100\,\mathrm{Mm^{-1}}$ (a typical value for Leipzig, Germany, in central Europe) in extreme situations. The extinction minimum was given at 4-5 km with values close to $1\,\mathrm{Mm^{-1}}$. Higher up, the UTLS wildfire smoke
caused a re-increase in the particle extinction values. Stable atmospheric conditions with a low amount of precipitation and correspondingly weak removal of particles by ice-phase cloud scavenging and cloud-related deposition processes favors long range transport of aerosol pollution from the industrial centers in the northern hemisphere towards the High Arctic during winter (Browse et al., 2012). Removal of aerosol pollution by dry deposition (caused by downward mixing of particles and removal at the surface) is also low in winter over the snow and ice-covered regions (Willis et al., 2019). The less well-defined
extinction profile structures observed from March to May 2020 in Fig. 7 occurred during the phase when the rather strong polar vortex weakened in March and collapsed after mid April 2020. The extremely strong polar vortex developed end of December 2019 and vanished completely not before the beginning of May 2020 (Ohneiser et al., 2021; Rinke et al., 2021).

During the summer months (June-August), aerosol layering is very different and the aerosol particle concentration especially in the lowest 1 km was roughly an order of magnitude lower than during the winter period. This finding is in full agreement
with the modeling study of Browse et al. (2012). They summarized that the seasonal cycle in Arctic aerosol is typified by high concentrations of large aged anthropogenic particles transported from lower latitudes in the late Arctic winter and early





spring followed by a sharp transition to low concentrations of locally sourced smaller particles in the summer. Wet scavenging processes control the seasonal variation in the aerosol conditions. Browse et al. (2012) show that the transition from high wintertime concentrations to low concentrations in the summer is controlled by the transition from ice-phase cloud scavenging
to the much more efficient warm cloud scavenging in the late spring troposphere. This seasonal cycle is amplified further by the appearance of warm drizzling cloud in the late spring and summer boundary layer. Low level liquid clouds and fog are ubiquitous in Arctic regions in summer and autumn. So, the seasonal cycle in Arctic aerosol, at least in the shallow ABL, is driven by temperature-dependent scavenging processes. The highest aerosol concentrations in summer were frequently found just above the shallow ABL as shown in Sect. 3.1. Stohl (2006) and Willis et al. (2018) pointed out that long-range transport of
aerosol in summer mainly occurs in the middle and upper troposphere. There is a lesser tendency for transport near the surface in summer than in winter.

Wildfire smoke (from North America and Siberia) occasionally filled again the upper troposphere in the summer of 2020. The increased extinction coefficients above 4 km height in June-July and August-September may have been caused by strong wildfires. Record-breaking smoke conditions as in the summer of 2019, however, did not occur in 2020.

In Fig. 8, we compare the MOSAiC winter (December to February) and summer (June to August) height profiles of the particle extinction coefficient with respective long-term (2006-2019) winter and summer profiles derived from polar observations (>65°N) with the spaceborne lidar CALIOP (Yang et al., 2021). The MOSAiC observations during the winter months 2019-2020 agree very well with the 15-year mean profile observations from space. In summer 2020, the lower troposphere up to 6 km height was obviously much cleaner than described by the 15-year mean CALIOP extinction values. The CALIOP data
include, e.g., the record-breaking smoke summer of 2019 and further smoke-polluted summers occurring since 2010 (Xian et al., 2022b).

We used the opportunity to compare the summer profiles in Fig. 8 with vertically resolved aerosol observations by means of a tethered balloon system (TBS) during the two summer periods of 2017 and 2018 (Creamean et al., 2021). The TBS was deployed at the U.S. Department of Energy Atmospheric Radiation Measurement Program's facility at Oliktok Point,
Alaska (70.51°N, 149.86°W), about 250 km southeast of the well-known AERONET station of Utqiagvik (Barrows, 71.3°N, 156.7°W). Based on 176 profiles of the vertical distribution of the particle number concentration (all particles with diameter >140 nm, $n_{70,\mathrm{dry}}$) in the summer seasons (June-August) of 2017 and 2018, the vertical distribution of the Arctic aerosol was characterized up to 1500 m. For this comparison, we converted the extinction coefficients in Fig. 8 into particle number concentrations $n_{50,\mathrm{dry}}$ for Arctic particles as described in Sect. 2.6. Good agreement between the MOSAiC, CALIOP, and
TBS aerosol profiles were found. The lidar-derived values of $n_{50,\mathrm{dry}}$ were 15-30 cm$^{-3}$ (MOSAiC, summer 2020) and around 40-50 cm$^{-3}$ (CALIOP, 2006-2019 summer mean) at 1000-1500 m height. The in situ observations of Creamean et al. (2021) showed values of $n_{70,\mathrm{dry}}$ of 60±15 cm$^{-3}$ close to the ground in northern Alaska and 30±10 cm$^{-3}$ at the base of frequently occurring cloud layers. Typical cloud base heights are around 1000±500 m during the summer months (Shupe et al., 2011; Creamean et al., 2021).



## 3.3 MOSAiC annual cycle: Aerosol optical thickness

Numerous reports on the Arctic aerosol conditions are available based on sunphotometer observations of the spectral aerosol optical thickness (AOT) (e.g., Tomasi et al., 2012, 2015; Xian et al., 2022a). However, these aerosol characterization studies are restricted to the sunlight period. Figure 9 now shows the AOT annual cycle for the entire MOSAiC year (2019-2020), derived from the lidar observations that are not restricted to the summer halfyear. Several AOT time series for different vertical columns are presented. The AOTs were calculated from the monthly mean height profiles of the extinction coefficient. In contrast to Fig. 7, we used a lidar ratio of 55 sr in the conversion of backscatter to extinction coefficients for heights <5 km only. For the heights above >5 km, we used a smoke lidar ratio of 85 sr (Ohneiser et al., 2021). We further assumed that the backscatter coefficient at the minimum measurement height of about 100 m represents the backscatter conditions at the surface as well. In this approach, we ignore the clean marine aerosol impact in the summer Arctic boundary layer on the AOT computation. Thus the AOT for the months from June-September may be slightly overestimated by about 0.002-0.005.

According to the 2006-2019 CALIOP observations for the Arctic area >65°N (Yang et al., 2021) in Fig. 9, the lower tropospheric AOT (0-5 km height) shows an annual cycle with a maximum in winter and a minimum in summer. CALIOP well detects the backscatter from the lower troposphere up to 5 km height, but is not very sensitive to weak backscatter contributions from the upper troposphere and lower stratosphere. Undetected aerosol contributions to AOT are typically of the order of 0.03 at 532 nm according to studies of Kim et al. (2017) and Toth et al. (2018). This bias explains roughly the difference between the MOSAiC AOTs (for the 0-10 km height range) and the CALIOP AOTs (for the total vertical column) in the summer of 2020 (June-July and August-September 2020), when the strong Siberian smoke layer in the UTLS height range was no longer present. The MOSAiC AOT summer values (0-10 km) are in good agreement with respective long-term Arctic AERONET observations at Thule and Ittoqqortoormiit. In summer, the long-term mean 500 nm AOT is 0.06 to 0.07 at Thule (76.5°N, 68.7°W) and Ittoqqortoormiit (latitude 70.5°N, 22°W) (Xian et al., 2022a).

All in all, the aerosol layering situation was not common in the MOSAiC year because of the UTLS aerosol. Even the AOT for the lowest 5 km over Polarstern was affected by descending and downward mixed smoke, especially during the months with the collapsing and vanishing polar vortex (April and May 2020). The deviation between the CALIOP AOT and the MOSAiC AOT (0-5 km) in April and May 2020 in Fig. 9 corroborates this hypothesis. The tropospheric AOT (up to 10 km) was generally strongly affected by Siberian smoke until May 2020. The UTLS AOT (5-20 km) was 0.08 (October 2019) to 0.04 (May 2020).

## 4 MOSAiC time series of cloud-relevant aerosol properties

In this section, we present our lidar results in terms of time series of CCN and INP concentration estimates at different height levels. We include the MOSAiC in situ observations of the particle number concentrations $n_{50,\mathrm{dry}}$ (Boyer et al., 2023) and of ice nucleating particles $n_{\mathrm{INP}}$ (Creamean et al., 2022) aboard Polarstern into this discussion. The MOSAiC CCN in situ observations presented by Dada et al. (2022) are considered as well.



### 4.1 CCN concentration at the surface, 250 m, and 2000 m height

In Fig. 10, the lidar-derived time series of $n_{50,\mathrm{dry}}$, i.e., of $n_{\mathrm{CCN}}$ for a supersaturation of 0.2% at 250 m and 2000 m height and the monthly means of $n_{50,\mathrm{dry}}$ measured in situ aboard Polarstern are shown. As mentioned, we selected a near-surface lidar height (250 m above sea level) and the height of 2000 m to show aerosol conditions relevant for the formation of low-level

clouds and stratiform mixed-phase clouds in the lower free troposphere, respectively. Peng et al. (2023) showed that ABL top height was most of the time around 200 m over the Polarstern during the MOSAiC year. Each lidar data point in Fig. 10 represents a several-hour observation (signal averaging time). To minimize the impact from foggy conditions, we considered lidar observations with a 532 nm backscatter coefficient of $<1$ $\mathrm{Mm}^{-1}$ $\mathrm{sr}^{-1}$ or extinction coefficients $<55$ $\mathrm{Mm}^{-1}$. Thus, after conversion of the extinction coefficients, only $n_{50,\mathrm{dry}}$ values $<700$ $\mathrm{cm}^{-3}$ remained. Gaps in the lidar time series indicate days

at which lidar observations were not possible because of fog and low-level-clouds.

In accordance with the observations of optical properties, strong differences in the CCN concentration between winter and summer are found at 250 m height in Fig. 10. In the lowest part of the troposphere, the $n_{50,\mathrm{dry}}$ or $n_{\mathrm{CCN}}$ values were mostly $>100$ $\mathrm{cm}^{-3}$ and sometimes up to 500-600 $\mathrm{cm}^{-3}$ in the period from November 2019 to April 2020. They were $<40$ $\mathrm{cm}^{-3}$ during the summer months when marine CCNs dominate. Such a strong contrast between winter and summer is not found for

the 2000 m height level. Here, the impact of continental aerosols dominate the CCN concentration levels throughout the year.

According to the in situ CCN observations aboard Polarstern (Dada et al., 2022), the background aerosol CCN values (for a supersaturation level of 0.2-0.3%) increased from $<50$ $\mathrm{cm}^{-3}$ October-December 2019, to about 100 $\mathrm{cm}^{-3}$ in Januray-March 2020, and 100-200 $\mathrm{cm}^{-3}$ in April and the first half of May 2020. Many short-term CCN concentration peaks around 200-300 $\mathrm{cm}^{-3}$ (November-December), 400-550 $\mathrm{cm}^{-3}$ (January-February) and even 650 $\mathrm{cm}^{-3}$ (April 2020) were measured aboard

Polarstern. Similar features (increasing values with time) are visible in the lidar observations at 250 m height in Fig. 10.

The in situ measured $n_{50,\mathrm{dry}}$ values (considering particles with diameters from 100-500 nm) were, on average, lower than the lidar estimates during the winter period (October-April). However, the monthly mean median particle number concentrations (not shown) were, as the lidar values, also clearly $>100$ $\mathrm{cm}^{-3}$ in the months of January, February, April and May (Boyer et al., 2023). The in situ observations also showed particle number concentrations $<40$ $\mathrm{cm}^{-3}$ in summer in terms of monthly median

values.

It is noteworthy to mention in this context, that the winter deviations between the surface in situ and lidar observations at 250 m may be partly related to the occurrence of low-level jets (LLJs) (López-García et al., 2022). The LLJ is defined as the local maximum in the vertical profile of wind speed in the lower troposphere, i.e., the wind speed is lower below and above the LLJ height. LLJs were found in about 50-60% of all MOSAiC radiosonde wind profiles. In about 20-30% out of all cases the

LLJ height was below 250 m height. It can be expected that dry deposition (removal) of particles is higher below the LLJ height and higher for air masses traveling above the LLJ height. The ABL top height was most of the time around 200 m height, and even below 200 m especially during the winter months Peng et al. (2023).

The MOSAiC observations are in good agreement with other measurements in remote areas at high latitudes far away from centers of anthropogenic haze. Tatzelt et al. (2022) presented shipborne in situ measurements of CCN concentrations conducted





in the Southern Ocean during the Antarctic Circumnavigation Expedition (ACE) from December 2016 to March 2017 (summer season). They found mostly CCN values of 50-200 cm$^{-3}$ for 0.2% supersaturation, but sometimes also more than 500 cm$^{-3}$ or less than 5 cm$^{-3}$. Herenz et al. (2018) and Chang et al. (2022) performed observations of CCN concentrations in the Canadian Arctic in May 2014 and July-August 2016, respectively, and found CCN concentrations mostly from 20-150 cm$^{-3}$ (Herenz et al., 2018) and 20-80 cm$^{-3}$ (Chang et al., 2022). Hartmann et al. (2021) reported CCN concentrations mostly from almost

zero to 250 cm$^{-3}$ ($S_{\mathrm{WAT}} = 1.002$) in the European Arctic at latitudes up to 83.7°N in May–July 2017.

## 4.2   INP concentration at the surface, 250 m, and 2000 m height

Figure 11 shows the lidar results in terms of estimates for the INP concentration. The same data set of particle backscatter coefficients as used in the foregoing CCN section is considered here. Now, the measured optical properties are converted into particle surface area concentrations which are input in the INP estimation (Sect. 2.8). The Arctic aerosol model (providing the

particle surface area concentration) is used to calculate the wintertime INP series (Ullrich et al., 2017) at 250 and 2000 m for a typical ice-nucleation temperature of −25°C, and also at 2000 m in summer for realistic ice-nucleation temperature of −15°C assuming that most mixed-phase clouds are in the lowermost 2.5 km of the atmosphere. The SSA INP parameterization of Alpert et al. (2022) is used in the case of the lidar INP estimates at 250 m in summer assuming a pure marine environment in the shallow ABL.

In winter, we assume that the only soil dust particles are able to trigger ice nucleation. We have adjusted the dust fraction to 1% (fractional contribution to the particle surface area concentration) in Fig. 11. For 1% dust, the lidar INP estimates at 250 m are in good agreement with the surface observations (Creamean et al., 2022) which served as a reference or guide to adjust our INP estimation. The impact of 1% pure dust is equivalent to 10% coated dust particles (with an order of magnitude lower ice-nucleating efficiency). In summer, we assume that continental aerosol particles (and thus dust particles) are absent in the

Arctic ABL so that the aerosol in the lowermost tropospheric layer is of local marine origin. Good agreement between the lidar estimates at 250 m and the in situ measured INP concentration is obtained without any adjustment of the lidar estimation.

The strong difference between the winter and summer INP levels is again visible and to a large part the result of the effective wet removal of continental aerosol during long range transport to the central Arctic in summer (Browse et al., 2012). Furthermore, the assumed higher ice-nucleation temperatures in summer (−10° to −15°C) contribute to the strong winter-

to-summer contrast. The INP concentration increases by roughly an order of magnitude with 5 K temperature decrease. The temperature effect is best visible in the INP time series for the 2000 m height level where the continental aerosol (and thus soil dust) is assumed to control INP conditions throughout the year, but ice-nucleation temperatures are assumed to be about 10 K higher in summer.

In the winter ABL, the INP concentration was mostly found between 0.01-0.1 L$^{-1}$ for the ice-nucleation temperature of

−25°C, while in summer, the ABL INP values ranged from 0.00001-0.001 L$^{-1}$ for the ice-nucleation temperature of −10°C, and were thus 2-3 orders of magnitude lower than the winter ABL INP values. However, the marine INPs of biogenic origin are much more efficient INPs at temperatures around −10°C than mineral dust INPs so that significant ice formation is possible as the study of Griesche et al. (2021) documented. In the summer ABL, a much higher amount of ice-containing clouds at



temperatures $> -10°C$ was counted in the Arctic in the summer of 2017 as, at the same time, in the free troposphere at these
likewise high ice-nucleation temperatures.

The hypothesis that biogenic INPs dominated the INP concentrations in summer is also corroborated by the detailed
temperature-dependent INP measurements aboard Polarstern (Creamean et al., 2022). The authors showed in situ measured
time series of $n_{\mathrm{INP}}$ for temperatures of $-10, -12.5, -15, -20, -22.5,$ and $-25°C$ from October 2019 to September 2020. And
only during the summer months (June to August), INPs were observed for high temperatures of $-10$ and $-12.5°C$. In winter,
the INP concentrations were close to zero for these high temperatures because dust particles are not ice-active at temperatures
$> -15°C$.

The MOSAiC observations are in good agreement with other INP measurements at high latitudes, far away from strong
sources of pollution. Tatzelt et al. (2022) also presented shipborne observation of INP concentrations conducted in the Southern
Ocean during ACE and found a strong accumulation of values within 0.05-0.1 $L^{-1}$ (interquartile range) for the temperature of
$-25°C$. Observations at Ny-Ålesund (78.9°N, 11.9°E) in Svalbard, Norway, in October-November 2019 and March-April 2020
yielded INP concentrations mostly in the range from 0.13-0.3 $L^{-1}$ (interquartile range) from 6 October to 15 November 2019
and from 0.2-0.55 $L^{-1}$ from in 16 March to 22 April 2020 for the temperature of $-25°C$. The Polarstern was more than 500 km
north of Ny-Ålesund during the winter period until April 2020. Si et al. (2019) reported INP concentrations accumulating from
0.04-0.4 $L^{-1}$ for $-25°C$, measured in the Canadian High Arctic (82.5°N, 62.5°W) during March 2016. Hartmann et al. (2021)
found INP values of 0.03-2 $L^{-1}$ for $-25°C$ during a Polarstern cruise in the European Arctic up to 83.7°N in May-July 2017.
Finally Sze et al. (2022) analyzed two-year-long INP measurements (from July 2018 to September 2020) at Villum 5 Research
Station in Northern Greenland (81.6°N, 16.7°W). The observations suggest INP concentrations mainly from 0.03-0.7 $L^{-1}$ at
$-25°C$. A clear indication for the dominance of biogenic INPs during the summer months was highlighted.

## 4.3 INP concentration close to the tropopause

During the MOSAiC winter halfyear (October 2019 to April 2020), aged Siberian wildfire smoke prevailed in the upper
troposphere over the High Arctic (Ohneiser et al., 2021). The smoke layer polluted the height range from about 5 km to about
20 km height as shown in Fig. 6, and thus the uppermost 3-4 km of the troposphere as shown in Fig. 7. The MOSAiC lidar
and radar observations aboard Polarstern offer a unique opportunity to investigate the impact of aged wildfire smoke on cirrus
formation. A first case study, i.e., a first $n_{\mathrm{INP}}$ vs $n_{\mathrm{ICE}}$ closure study, was discussed in Engelmann et al. (2021). Smoke particles
are moderately efficient INPs at temperatures from $-50°$ to $-70°C$.

Figure 12 shows the MOSAiC time series of smoke INP estimates close to the tropopause from October 2019 to the beginning
of May 2020, and then again in September 2020. In addition, dust INP estimates for the summer months from June to August
2020 are included in the figure assuming a mixture of smoke, dust, and aged anthropogenic haze in the upper troposphere as
indicated in Fig. 3 for the 5 August 2020. The assumed 10% dust fraction is in line with the CALIOP observations for the polar
region presented by Yang et al. (2021). The dust and smoke INP retrieval schemes were explained in Sect. 2.8.3. Each lidar
value is based on lidar observations over several hours (signal averaging period) (Ohneiser et al., 2021).



As can be seen in Fig. 12, for the selected conditions ($-65°$C, smoke during winter halfyear, 10% dust fraction in summer, $S_{ICE} = 1.42$ in the case of smoke, $S_{ICE} = 1.25$ in the case of dust), INP numbers accumulate in the 1-20 L$^{-1}$ range. Note, that the smoke INP data set (green triangles) can entirely be reproduced by dust INP values (assuming a 10% dust contribution to
the lidar-derived particle surface area concentration) and assuming an $S_{ICE}$ of 1.25. However, all our multiwavelength Raman lidar products covering the winter MOSAiC months from October 2019 to April 2020 clearly point to smoke aerosol (Ohneiser et al., 2021). Disregarding this discussion, INP concentrations of 1-20 L$^{-1}$ as shown in Fig. 12 are sufficient to initiate significant ice production via heterogeneous ice nucleation and to widely suppress homogeneous freezing. For homogeneous ice nucleation a supersaturation of $S_{ICE} = 1.5$ is required at $-65°$C. These INP numbers of 1-20 L$^{-1}$ are in consistency with
MOSAiC lidar-radar-based retrievals of ice crystal number concentrations $n_{ICE}$ in ice virga below the freshly formed cirrus cloud decks by using the method presented by Bühl et al. (2019) and Ansmann et al. (2019b) and applied in Engelmann et al. (2021). Meanwhile we analyzed about 10 MOSAiC cirrus systems that evolved in the smoke-polluted upper troposphere over Polarstern in the winter of 2019-2020.

The final Fig. 13 shows an example of the impact of wildfire smoke on cirrus formation. Polarstern was at 88°N. Four days
of continuous lidar observations from 25-29 February 2020 are presented.The smoke layer is clearly visible as yellow layer around 10 km height. Heterogeneous ice nucleation occurred in the yellow smoke layer at temperatures from $-69$ to $-73°$C and RH values were mostly between 65% and 72% in the height range from 9-10 km on 25-28 Februar 2020 according to the MOSAiC radiosonde temperature and RH observations. Frequently occurring gravity waves, typically causing updraft speeds around 20 cm/s (Barahona et al., 2019), probably lofted the humid air parcels so that RH increases with decreasing temperature
during upward motion and the required ice supersaturation threshold of 1.35-1.45 was reached and finally exceeded and ice nucleation on the smoke particles began. Immediately after nucleation, the ice crystals grew fast by water vapor deposition on the crystals and started to fall out of the smoke layer. They formed long virga, partly visible down to heights of 6 km in Fig. 13. Below 6 km height, the air was dry and the crystals evaporated.

## 5   Conclusion/Outlook

The multiwavelength polarization Raman lidar Polly aboard Polarstern continuously monitored Arctic aerosols and clouds in the troposphere and stratosphere during the MOSAiC year from October 2019 to September 2020 (Engelmann et al., 2021; Ohneiser et al., 2021; Ansmann et al., 2022). As presented in this study, the lidar observations together with the in situ observations aboard Polarstern allowed a detailed insight into the vertical distributions of optical, microphysical, and cloud-relevant aerosol properties. A strong decrease of aerosol pollution with height was found during the winter months (October 2019 to
April 2020) up to about 4-5 km height. The aerosol concentration decreased by an order of magnitude within 2 km. The minimum in 4-5 km heights separated the Arctic haze in the lower atmosphere from wildfire smoke in the upper troposphere and lower stratosphere. In summer, rather clean conditions prevailed in the lowest 1 km, obviously a result of efficient wet removal of aerosol particles during long-range travel, and lofted continental aerosol plumes (containing anthropogenic haze, fire smoke,



and a small fraction of soil dust) from Europe, Asia and North America polluted the air from time to time, mostly above 1 km
height.

CCN and INP concentrations were estimated from the lidar observations by using lidar analysis methods developed during
recent years. The CCN concentration was found to strongly drop with height in winter in line with the observed decrease of
the aerosol backscatter and extinction coefficients. During summer, aerosol pollution in the ABL was, on average, an order
of magnitude lower than in winter. The MOSAiC measurements suggest that local marine particles prevailed in the Arctic
ABL over the Polarstern and that marine CCN and INP concentrations controlled low-level cloud processes. The time series
of the CCN concentration in the free troposphere did not show a pronounced annual cycle, because long-range transport of
continental particles permanently occurs during all seasons of the year.

While soil dust particles may by the main reservoir for INPs in winter and above the local ABL in summer, marine aerosol
(biogenic components) is most probably the main INP type in the summer Arctic ABL. As the MOSAiC observations reveal,
wildfires can no longer be ignored as an important source for Arctic aerosols (and thus INPs) in the upper troposphere, at cirrus
level. The observed INP concentration levels of 1-20 L$^{-1}$ close to the tropopause throughout the MOSAiC year suggested that
homogeneous ice nucleation was probably widely suppressed and thus of minor importance over the polar region in 2019-
2020. As an outlook, the MOSAiC observation provide an excellent data base to study aerosol-cloud interaction in the Arctic
atmosphere and especially smoke-cirrus interaction and also to test established and new INP parameterization schemes. The
results of these projects will be presented in follow-up articles.

## 6   Data availability

Polly lidar observations (level 0 data, measured signals) are in the PollyNet database (Polly, 2022). All the analysis products
are available at TROPOS upon request (polly@tropos.de) and at https://doi.pangaea.de/10.1594/PANGAEA.935539 (Ohneiser
et al., 2021). MOSAiC radiosonde data are available at https://doi.org/10.1594/PANGAEA.928656 (Maturilli et al., 2021)
Backward trajectory analysis has been performed by air mass transport computation with the NOAA (National Oceanic
and Atmospheric Administration) HYSPLIT (HYbrid Single-Particle Lagrangian Integrated Trajectory) model (HYSPLIT,
2022). AERONET and MICROPTOPS observational data are downloaded from the respoective data bases (AERONET, 2022;
AERONET-MAN, 2022).

## 7   Author contributions

The paper was written and designed by AA, KO and RE. The aerosol data analysis was performed by KO, RE, MR, JMC,
MCB, JB, CJ, and HGe. KO, RE, JMC, MCB, DAK, MR, PS, and UW were involved in the interpretation of the findings. RE,
HGr, MR, JH, and DA took care of the lidar observations aboard Polarstern during MOSAiC. SD and MM were responsible
for high-quality MOSAiC Polarstern radiosonde launches. All coauthors were actively involved in the extended discussions
and the elaboration of the final design of the manuscript



## 8 Competing interests

Daniel A. Knopf is a member of the editorial board of Atmospheric Chemistry and Physics

## 9 Financial support

The Multidisciplinary drifting Observatory for the Study of the Arctic Climate (MOSAiC) program was funded by the German Federal Ministry for Education and Research (BMBF) through financing the Alfred Wegener Institut Helmholtz Zentrum für Polar und Meeresforschung (AWI) and the Polarstern expedition PS122 under grant N-2014-H-060_Dethloff. The lidar analysis on smoke-cirrus interaction was further supported by BMBF funding of the SCiAMO project (MOSAIC-FKZ 03F0915A). The radiosonde program was funded by AWI awards AFMOSAiC-1_00 and AWI_PS122_00, the U.S. Department of Energy Atmospheric Radiation Measurement Program, and the German Weather Service. This project has also received funding from the European Union's Horizon 2020 research and innovation program ACTRIS-2 Integrating Activities (H2020-INFRAIA-2014 - 2015, grant agreement no. 654109). We gratefully acknowledge the funding by the Deutsche Forschungsgemeinschaft (DFG, German Research Foundation) – project no. 268020496 - TRR 172, within the Transregional Collaborative Research Center "ArctiC Amplification: Climate Relevant Atmospheric and SurfaCe Processes, and Feedback Mechanisms (AC)3". JMC acknowledges support by U.S. Department of Energy's (DOE) Atmospheric Radiation Mission (ARM) (grant no. DE-AC05-76RL01830 DE-SC0021034) and Atmospheric System Research (ASR) program (grant no. DE-SC0019745, DE-SC002204). DAK acknowledges support by U.S. Department of Energy's (DOE) Atmospheric System Research (ASR) program, Office of Biological and Environmental Research (OBER) (grant no. DE-SC0021034).

*Acknowledgements.* Data used in this article were produced as part of the international Multidisciplinary drifting Observatory for the Study of the Arctic Climate (MOSAiC) with the tag MOSAiC20192020 and the Project_ID: AWI_PS122_00. We would like to thank everyone who contributed to the measurements used here (Nixdorf et al., 2021). Radiosonde data were obtained through a partnership between the leading Alfred Wegener Institute, the Atmospheric Radiation Measurement user facility, a U.S. Department of Energy facility managed by the Biological and Environmental Research Program, and the German Weather Service (DWD). We would like to thank the RV Polarstern crew for their perfect logistical support during the one-year MOSAiC expedition.



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



**Table 1.** Overview of Polly observational products, used in this study, and exemplary (or typical) relative uncertainties in the determined and retrieved properties. Particle backscatter coefficients are measured at 355, 532, and 1064 nm, the other aerosol optical properties at 355 and 532 nm. $r$ denotes aerosol particle radius.

| Aerosol optical properties | Uncertainty |
|---|---|
| Backscatter coef. [Mm$^{-1}$ sr$^{-1}$] | $\leq$10% |
| Extinction coefficient [Mm$^{-1}$] | 20% |
| Lidar ratio [sr] | 25% |
| Depolarization ratio | $\leq$10% |
| Aerosol microphysical properties | |
| Volume conc. [$\mu$g m$^{-3}$] | $\leq$25% |
| Surface-area conc. [$\mu$m$^2$ cm$^{-3}$] | $\leq$25% |
| Number conc. ($r >$85 nm) [cm$^{-3}$] | 50% |
| Number conc. ($r >$290 nm) [cm$^{-3}$] | $\leq$25% |
| Cloud-relevant properties | |
| CCN conc. [cm$^{-3}$] | 50% |
| INP conc. [L$^{-1}$] | Order of magn. |

**Table 2.** Conversion parameters for Arctic aerosol, required in the conversion of the particle extinction coefficient $\sigma$ at 532 nm into particle number concentrations $n_{65}$, $n_{85}$, $n_{250}$, and $n_{290}$, surface area concentration $s$, and volume concentration $v$. The mean values and range of mean values (from the 4 stations) for the conversion factors $c_v$, $c_s$, $c_{65}$, $c_{85}$, $c_{250}$, and $c_{290}$ are obtained from the extended AERONET data analysis (AERONET, 2022). The conversion factors are derived from the AERONET observations at Barrow (1997-2021), Thule (2007-2021), Pearl (2007-2019), and Kangerlussuaq (2008-2021). All conversion factors hold for 532 nm wavelength. The AERONET data analysis procedures applied to obtain the conversion factors are described in Mamouri and Ansmann (2016, 2017).

| Conversion factor | Value | Range of values |
|---|---|---|
| $c_v$ [10$^{-12}$ Mm] | 0.215 | 0.19-0.24 |
| $c_s$ [10$^{-12}$ Mm m$^2$ cm$^{-3}$] | 2.8 | 2.65-2.90 |
| $c_{65}$ [Mm cm$^{-3}$] | 12.5 | 11.2-15.0 |
| $c_{85}$ [Mm cm$^{-3}$] | 10.0 | 9.6-12.2 |
| $c_{250}$ [Mm cm$^{-3}$] | 0.25 | 0.22-0.28 |
| $c_{290}$ [Mm cm$^{-3}$] | 0.13 | 0.12-0.145 |



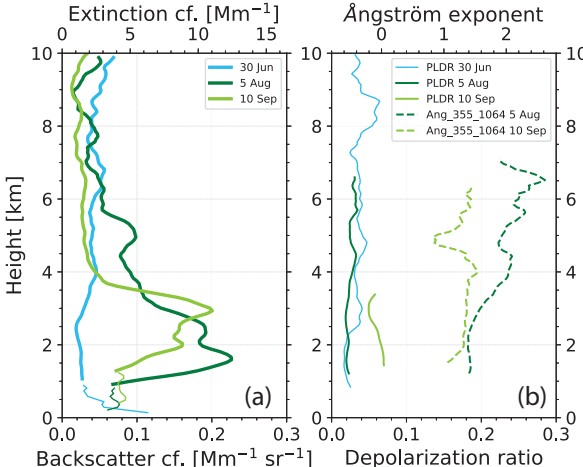

**Figure 1.** Pollution long-range transport towards the High Arctic at heights above 1 km observed with the Polarstern lidar on 5 August 2020 (lidar observations are averaged from 21:00-24:00 UTC, Polarstern position: 78.4°N, 6.0°W) and on 10 September 2020 (signal averaging from 18:15-21:10 UTC, Polarstern position: 88.7°N, 105.6°E). The measurement on 30 June 2020 shows clean background conditions (18:00-24:00 UTC, Polarstern at 81.8°N, 9.5°E). Backscatter and extinction profile segments from lidar observations with the near-range telescope are shown as dashed lines up to about 1 km height in (a). The 532 nm extinction coefficients are obtained by multiplying the backscatter coefficients with a lidar ratio of 55 sr. In (b), the particle linear depolarization ratio (PLDR) at 532 nm for all three days and the backscatter-related Ångström exponent (Ang, considering the backscatter coefficients at 355 and 1064 nm) for the two pollution events on 5 August and 10 September2020 are given. The Ångström exponent was 1.5-2.0 throughout the troposphere during the clean background conditions on 30 June (not shown).



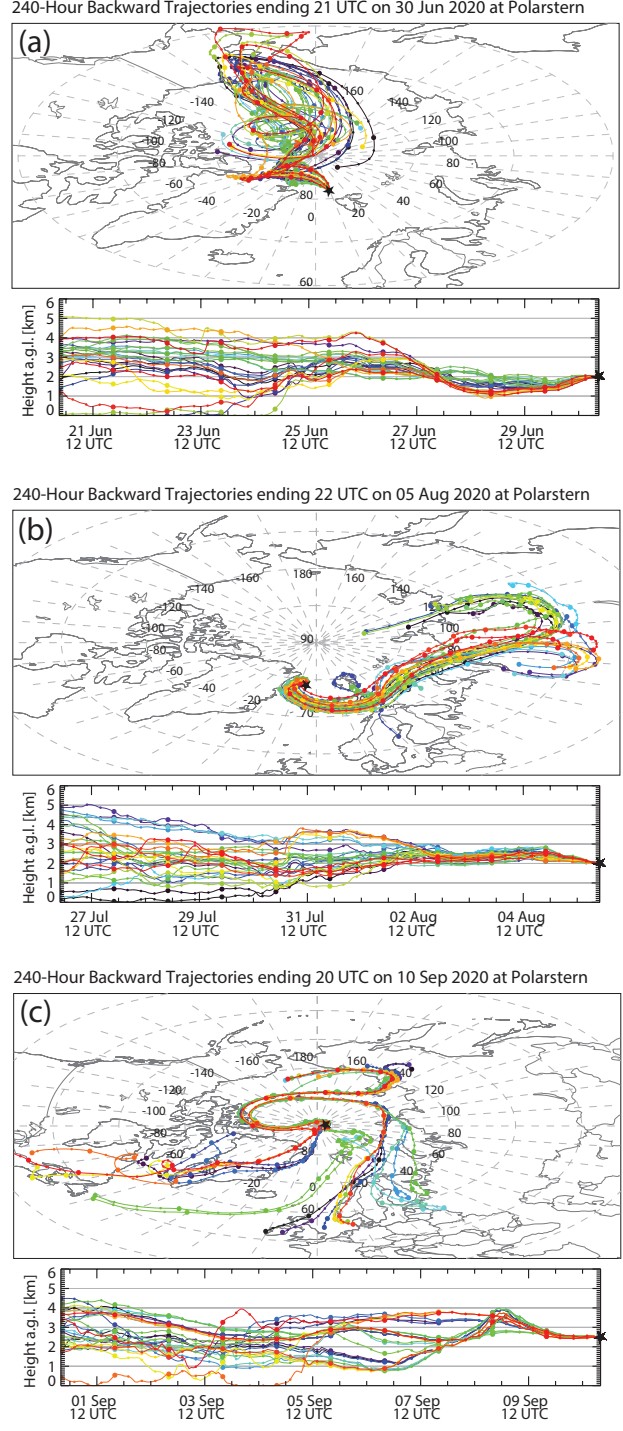

**Figure 2.** (a) HYSPLIT 10 d ensemble backward trajectories arriving over the Polarstern (indicated by a star) on (a) 30 June 2020, 21:00 UTC, (b) 5 August 2020, 22:00 UTC, and on (c) 10 September 2020, 21:00 UTC.



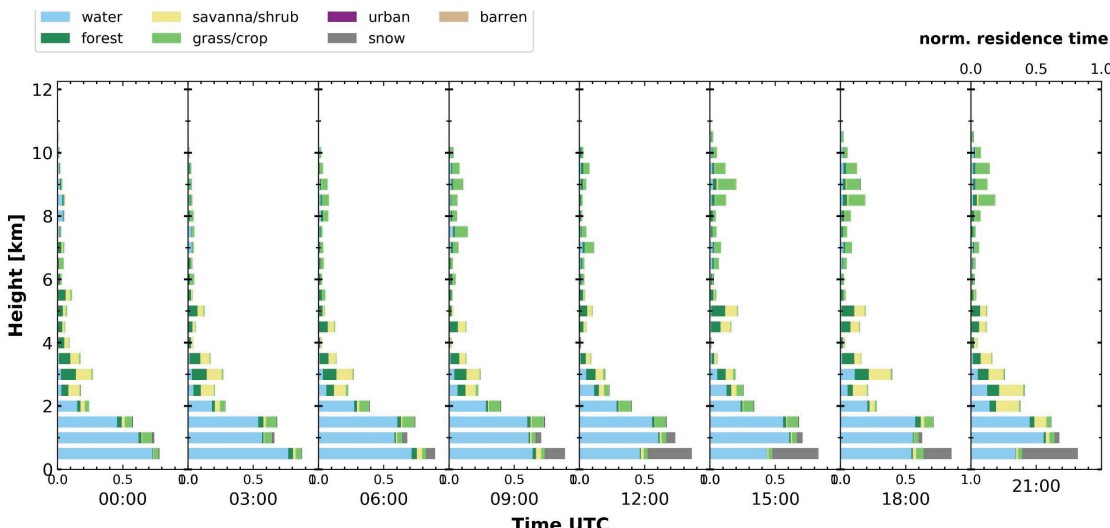

**Figure 3.** Vertically resolved air mass source attribution in 3 h intervals on 5 August 2020. The method of Radenz et al. (2021) is applied. The normalized (accumulated) residence time of air masses, when they traveled within the well-mixed boundary layer at heights below 2 km during the long-range transport, is given. The analysis is based on 10 d HYSPLIT backward trajectories arriving over Polarstern. The colors indicate different land cover classes. Continental particles contributed significantly to the backscatter and extinction coefficients, measured at heights >1 km on 5 August 2020, 18:00 and 21:00 UTC, shown in Fig. 1a.

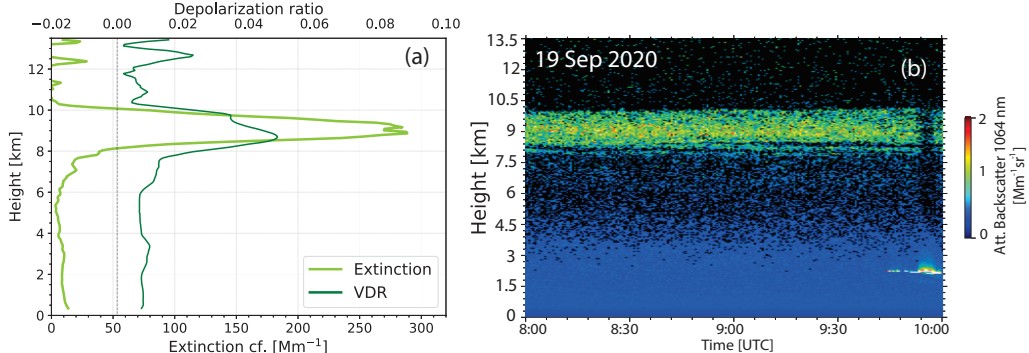

**Figure 4.** Wildfire smoke observed over Polarstern between 8 and 10 km height on 19 September 2020 (8:00-10:00 UTC, 89.1°N, 110°E). The smoke was probably lofted by pyroCb convection over northern parts of North America. Profiles of the 532 nm particle extinction coefficient (backscatter coefficient multiplied with a smome lidar ratio of 70 sr) and the volume depolarization ratio (VDR) are shown in (a). Mean profiles for the time period from 8:00-9:40 UTC are presented. In (b), the height-time display of the 1064 nm range-corrected signal (or attenuated backscatter coefficient), showing the 2 km thick smoke layer, is given. The volume depolarization ratio of 5% (smoke particle depolarization ratio of 6-7%) is indicative for fast pyroCb lofting. The 532 nm AOT of the smoke layer was about 0.4.





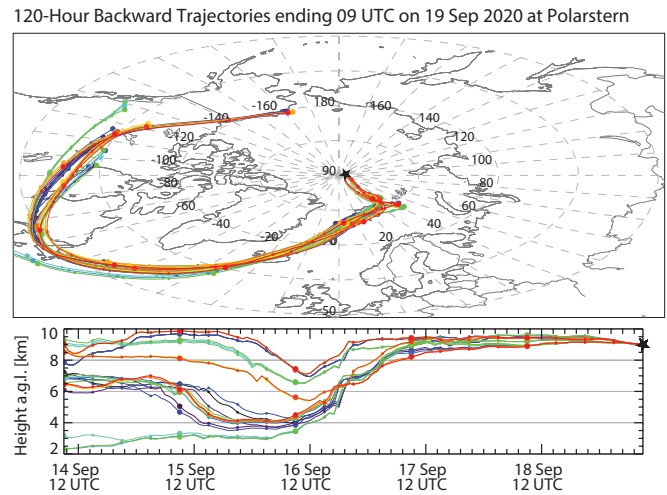

**Figure 5.** HYSPLIT 5 d ensemble backward trajectories arriving over the Polarstern (indicated by a star) on 19 September 2020, 09:00 UTC.

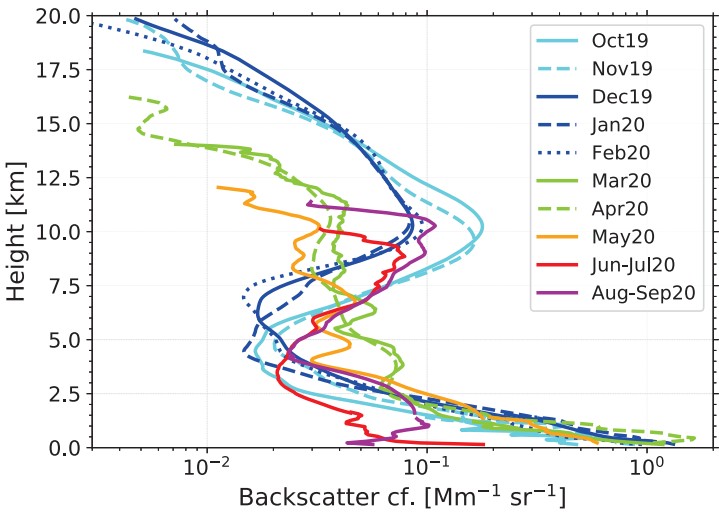

**Figure 6.** Aerosol layering over the High Arctic in 2019-2020. One-month and two-month mean particle backscatter profiles, measured at 532 nm, are shown. 9-15 daily observations per month are considered (October-March) and 5-8 per month during the summer months from June to September 2020. The UTLS height range (above 7.5 km) was strongly polluted by wildfire smoke (85% fraction) and Raikoke volcanic aerosol (15% fraction) during autumn and winter months (October 2019 - Februray 2020, cyan and blue colors).



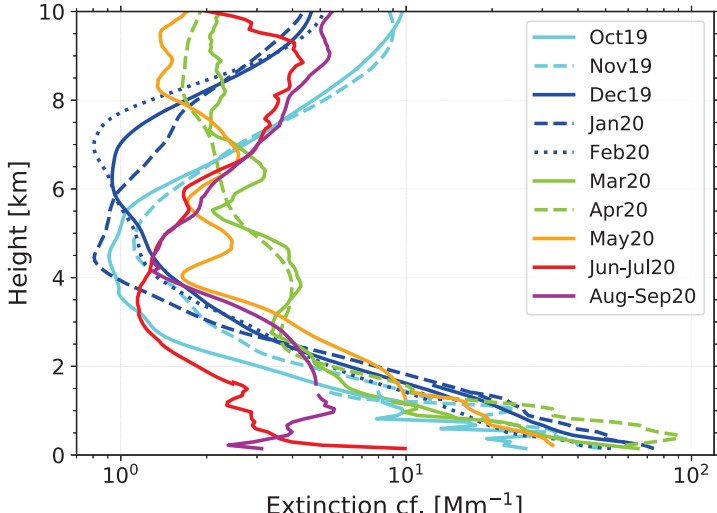

**Figure 7.** Tropospheric aerosol layering in terms of 1-month and 2-month mean particle light-extinction profiles (532 nm backsatter profiles shown in Fig. 6 times a lidar ratio of 55 sr). By combining lidar observations with the near-range telescope (covering the height range from 50-100 m up to 1.0-1.5 km) and the far-range telescope (covering the height range >1 km), particle extinction coefficients for the entire vertical tropospheric column could be determined. Continental aerosol pollution, soil dust, and biomass burning smoke dominated the aerosol conditions in the lowest 5 km, while pyroCb-lofted and self-lofted wildfire smoke causes the re-increase of the extinction values at heights >5 km (October 2019 - February 2020, June - September 2020).



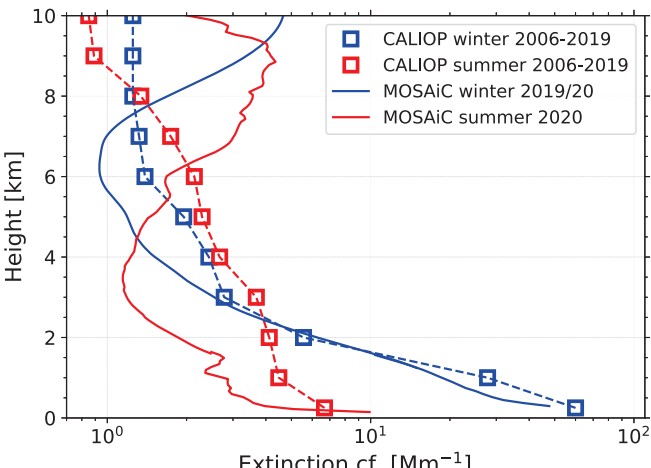

**Figure 8.** CALIOP (2006-2019) vs. MOSAiC (2019-2020) seasonal mean particle extinction profiles (532 nm) for the winter season (December-February) and summer season (June-August). CALIOP profiles are taken from Fig. 6 in Yang et al. (2021) and normalized with AOT shown in Fig. 3 in Yang et al. (2021). All CALIOP observations performed at latitudes >65°N are considered. The MOSAiC extinction profiles are again computed from the 532 nm backscatter profiles (multiplied by a lidar ratio of 55 sr).

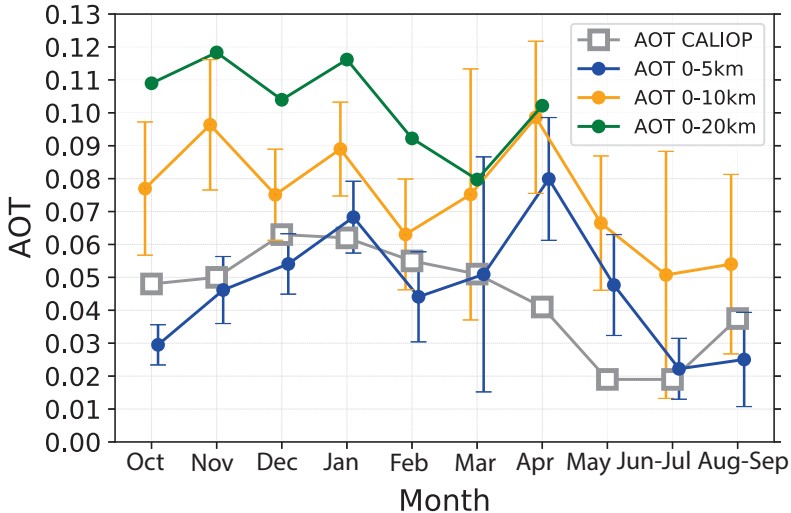

**Figure 9.** One-month and two-month mean AOTs for different height ranges measured during the MOSAiC expedition (October 2019 to September 2020). Backscatter profiles (532 nm) were multiplied by a typical tropospheric lidar ratio of 55 sr (0-5 km height) and a smoke lidar ratio of 85 sr (5-20 km) before the AOTs were computed. CALIOP AOT values (2006-2019 monthly means, 65-82°N mean, 0-12 km height range) are from Fig. 3. in Yang et al. (2021).



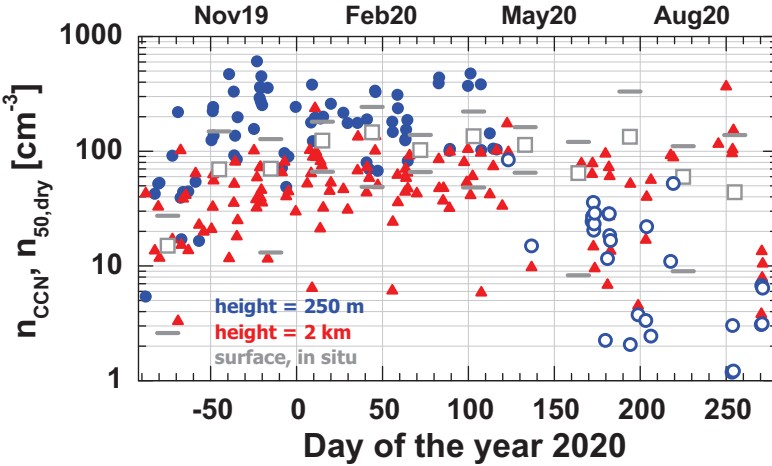

**Figure 10.** Annual cycle of the CCN concentration (0.2% supersaturation) during the MOSAiC year as observed in situ aboard Polarstern (gray squares, monthly mean CCN values, SD indicated by short grey horizontal lines) (Boyer et al., 2023) and estimated from Polarstern lidar observations at 250 m (blue circles) and 2000 m height (red triangles). Only $n_{CCN} < 700$ cm$^{-3}$ (for the 250 m height level) are considered, corresponding to cases with extinction coefficients $<55$ Mm$^{-1}$ (non-foggy conditions).

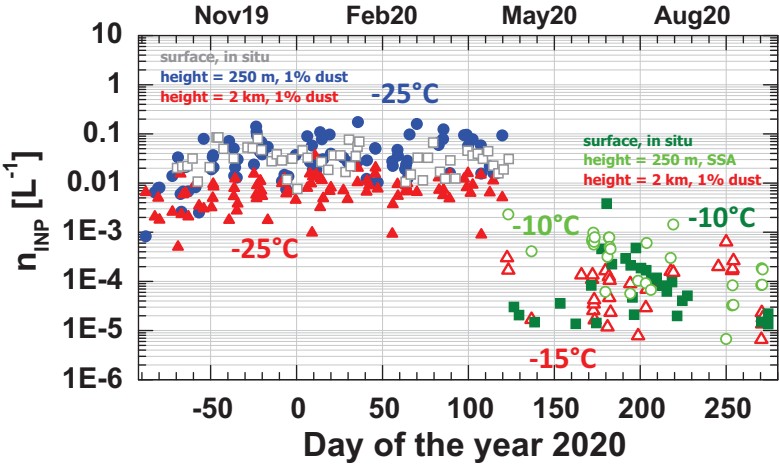

**Figure 11.** Annual cycle of the INP concentration during the MOSAiC year as observed in situ aboard Polarstern (gray and dark green squares, daily mean INP values) (Creamean et al., 2022) and estimated from Polarstern lidar observations at 250 m (blue and light green circles) and 2000 m height (red triangles) for ice-nucleating temperatures of $-25$°C in winter and $-10$°C (surface, 250 m) and $-15$°C (2000 m) in summer. Soil dust (1% contribution to the aerosol surface area concentration) is assumed to be the only ice-active aerosol type at 250 m (in winter) and 2000 m (in winter and summer), while sea spray aerosol is assumed to be the only INP type at 250 m in summer.



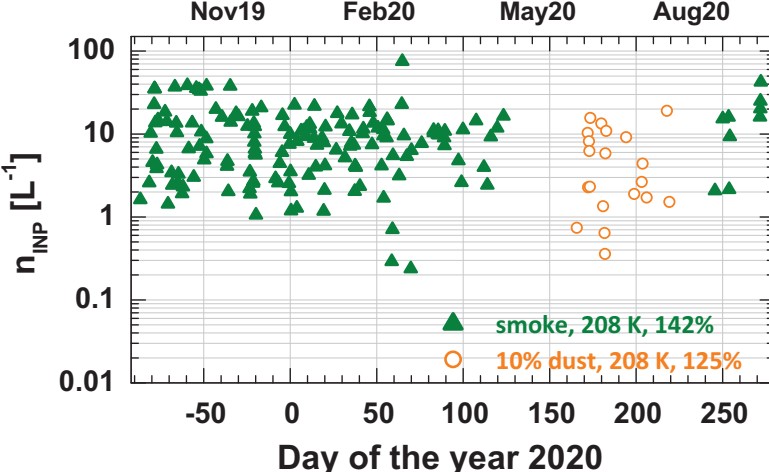

**Figure 12.** Lidar estimates of smoke INP concentrations (green triangles, October 2019 to September 2020, immersion freezing mode) for the height level of 1 km below the tropopause. In addition, estimated dust INP concentrations (orange circles, deposition ice nucleation mode, 10% dust fraction) are shown. In all computations, a temperature of $-65°$C is assumed and an ice supersaturation $S_{ICE}$ of 1.25 (dust INPs) and 1.42 (smoke INPs).

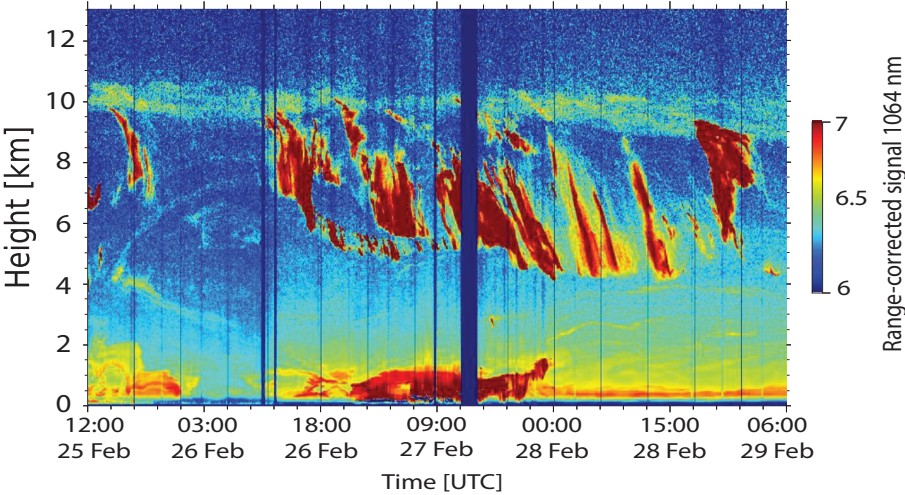

**Figure 13.** Lidar observations of cirrus formation in a wildfire smoke layer (in yellow around 10 km) on 25-29 February 2020. Coherent fall strikes (virga in orange and red above 4 km) of fastly growing, falling ice crystals developed quickly after nucleation of ice crystals in the smoke layer. The virga reached down to almost 4 km where the crystals evaporated in dry air. Temperatures were close to $-70°$C at cirrus formation level. The range-corrected 1064 nm lidar return signal is shown in logarithmic scales (arbitray units).