# Peer review of "Annual cycle of aerosol properties over the central Arctic during MOSAiC 2019-2020 — light-extinction, CCN, and INP levels from the boundary layer to the tropopause"

_EGUsphere, 2023_

## Referee Comment (RC1)

**Review comment on "Annual cycle of aerosol properties over the central Arctic during MOSAiC 2019-2020—light-extinction, CCN, and INP levels from the boundary layer to the tropopause" by Ansmann et al.**

**Anonymous Referee #1**

This paper presents the annual cycle of height-resolved aerosol properties measured over the central Arctic by the MOSAiC lidar complemented with in situ observations. This unique dataset provides insights into the vertical distribution of optical and cloud-relevant aerosol properties (CCN, INP) throughout the troposphere and different seasons, which is important for modeling clouds and the present and future Arctic climate. The analysis of the annual cycle was extended with several case studies, with an emphasis on the presence of a persistent wildfire smoke layer in the upper troposphere.

The paper is well written and has a clear structure. I recommend publication after addressing the comments below.

**General comments**

1.  In Sect. 3.1 you present three case studies with clean and polluted conditions during summer before you discuss the annual cycle of the aerosol conditions during the MOSAiC year. The purpose of presenting these case studies is not yet clear to me. For example:
    - How did you select these case studies?
    - Is some information on the case studies required for the analysis of the annual cycle (e.g., derive lidar parameters that are used in the following sections)?

    Please extend the description of the analysis regarding the case studies (e.g., goal of the analysis, selection of the case studies, …).

2.  The detection of a pronounced and persistent wildfire smoke layer in the upper troposphere and lower stratosphere is an important MOSAiC highlight. However, I have the impression that the discussion about the smoke layer from previous studies was sometimes interrupting the flow. For example:
    - In line 474-475 you state that the annual cycle of the aerosol optical properties is shown in Fig. 6 and 7, so I was expecting that you would discuss the profiles of backscatter and extinction coefficients that you observed during MOSAiC. However, in the following lines you don't focus on the discussion of these figures/your observations, but you report the findings regarding the smoke layer from previous studies (see also specific comment 14).
    - In line 52-63 you put the discussion of the smoke layer together with the description of the instrument setup aboard Polarstern, which does not really fit.

    I fully agree that the presence of the persistent wildfire smoke layer is an interesting feature/finding that needs to be discussed, but please revise if the structure of some paragraphs could be revised to maintain a clear flow.

**Specific comments**

1. Line 113-118: This paragraph seems a bit out of place. You already introduced in the previous paragraph that local INPs of biogenic origin seem to control ice nucleation in the boundary layer in summer (line 101-102). Also, the definition of homogeneous and heterogeneous nucleation should be moved forward in the paper (before discussing the results of different INP studies). Please restructure the paragraphs regarding INPs. You could also add reference to Carlsen and David (2022) and Sze et al. (2023) as additional motivation for the importance of INP on cloud phase. For example, Carlsen and David (2022) saw a clear relationship between cloud phase over open ocean, snow and sea ice cover suggesting that local INPs are important for cloud phase.
Carlsen, T., & David, R. O. (2022). Spaceborne evidence that ice-nucleating particles influence high-latitude cloud phase. Geophysical Research Letters, 49, e2022GL098041. https://doi.org/10.1029/2022GL098041
Sze, K. C. H., Wex, H., Hartmann, M., Skov, H., Massling, A., Villanueva, D., and Stratmann, F.: Ice-nucleating particles in northern Greenland: annual cycles, biological contribution and parameterizations, Atmos. Chem. Phys., 23, 4741–4761, https://doi.org/10.5194/acp-23-4741-2023, 2023.

2. Line 165-173: In subsection 2.2 you describe the lidar instrument. Is there a reason why you only present the parameters measured/derived by the lidar in Sect. 2.5? I suggest presenting the parameters measured by the lidar already in Sect. 2.2. If you want to keep the parameters in a separate section, you should refer to section 2.5.

3. Line 175: In the introduction you mention that sun photometer measurements are possible from March to September (line 88-89). Why were no measurements performed between March and June aboard Polarstern? I think it would be nice if you could include the sun photometer measurements in Fig. 9 (see specific comment 19).

4. Line 182-185: In Sect. 2.1 you state that Polarstern was mainly drifting at latitudes > 85 °N. The CALIOP measurements published by Yang et al. (2021) were obtained at latitudes between 65 °N and 82 °C (between 2006 and 2019). Thus, the CALIOP and MOSAiC measurements do not cover the same region and time period. How does this influence the comparison between MOSAiC and CALIOP measurements (e.g., Fig. 8 and 9)? Please add a few sentences regarding this point.

5. Table 1: What is the difference between the term 'exemplary' and 'typical' uncertainty? I would suggest sticking to one term.

6. Line 299-301: Please add a reference.

7. Line 395-410: In Sect. 2.9 you describe the in situ measurements of aerosol microphysical properties and INP concentrations that were conducted on Polarstern. Maybe it makes sense to move Sect. 2.9 forward to the other instrumentation. Like that you first describe the applied instrumentation and then introduce the different data analysis methods.

8. Line 396-410: Please include some information on how the particle number, CCN, INP concentrations were measured (e.g., instrument type) aboard Polarstern.

9. Line 437: Where can I see the backward trajectories for the 4 km height? If they are not shown, please state that.

10. Fig. 1: In the caption of Fig. 1 you state that the observations with the near-range telescope are shown by the dashed line in Fig. 1.a. This is not the case. The near-range observations are represented by thinner lines. Please change the plot or caption accordingly.

11. Fig. 3: Comments to Fig. 3:
    a. The 0 and 1 labels on the x-axis overlap. Please revise the labelling.
    b. For the right panel you show the residence time on the upper x-axis, while for the other panels you show the residence time on the lower x-axis. I suggest showing the residence time on the upper x-axis for all panels and the interval time on the bottom x-axis.

12. Line 475-486: The aim of this section is to present the annual cycle of the optical aerosol properties measured during the MOSAiC year (Fig. 6 and Fig. 7). However, in the first paragraph you focus mainly on the pronounced wildfire smoke layer that was discussed in detail by previous studies. I would suggest focusing on your observation and presenting Fig. 6 in detail before you discuss the results of other studies (see also general comment 2). For example, you could highlight the presence of the smoke layer in the upper troposphere (especially in winter) in Fig. 6.

13. Fig. 6: In the caption of Fig. 6 you state that 9-15 (October-March)/5-8 (June-September) daily observations per month were considered. Are those all measurements that were available or according to which criteria did you chose the observations? Please specify. How many observations were considered for April and May? Please add the number of observations also for April and May. Alternatively, you can specify the number of observations for each one-month/two-month period in brackets in the legend of the figure.

14. Line 532-533: Here you state that the MOSAiC and CALIOP observations agree well during the winter months (Fig. 8). However, above 7.5 km, the extinction coefficients measured during MOSAiC are higher compared to the CALIOP measurements, which is an effect of the smoke layer. You could include this in the interpretation of the figure. The same holds true for the extinction coefficients measured in the upper troposphere during the summer season.

15. Fig. 8: Comments to Fig. 8:
    a. I think it would be beneficial if you could include the standard deviation for the CALIOP measurements. This allows to assess if the MOSAiC year was significantly different compared to the 15-year mean CALIOP profile.
    b. To support the comparison between the 15-year CALIOP and MOSAiC lidar measurements, you could investigate if the MOSAiC and CALIOP measurements of the same time period are in good agreement: I.e. can the CALIOP profile from October 2019 to September 2020 capture the vertical profile measured by MOSAiC?

16. Line 537-549: In this paragraph you compare height-resolved aerosol observations from MOSAiC, a TBS and CALIOP. However, these observations were obtained in different regions and over different time periods, which makes a direct comparison difficult. What is the goal of this comparison? In my opinion it does not make sense to compare the aerosol observations of these three datasets, as they are quite different. As this paragraph is not required for the outline of the paper, I suggest removing this paragraph. If you decide to keep this comparison in the final manuscript, you should include a figure to illustrate the comparison. Also, I am a bit surprised that you state that good agreement between the MOSAiC, CALIOP and TBS

aerosol profiles was found (line 544-545), when you write in line 533-534 that in summer, the lower troposphere measured during MOSAiC was much cleaner than described by the 15-year mean CALIOP profiles.

17. Fig. 9: Comments to Fig. 9:
    a. In Sect. 2.3 you introduce the sun photometer measurements and state that these measurements were performed between June to September. I think it would be beneficial to include the sun photometer measurements in Fig. 9.
    b. Please include the definition of the vertical bar in the caption (uncertainty?). Why is there no vertical range for AOT 0-20 km? Why are there no data points between May and September for AOT 0-20 km? Please revise.

18. Fig. 10: Comments to Fig. 10:
    a. The lower standard deviation bar is missing for some values. Please include it.
    b. What is the difference between the filled/empty blue circle? Please specify.

19. Line 637-650: In this section you describe the strong difference between the winter and summer INP levels that is visible in Fig. 11. This pronounced annual cycle in the INP concentration can be mainly explained by the difference in summer/winter temperatures, as ice nucleation is strongly temperature dependent. To investigate the annual cycle of the INP concentration you should show the INP concentration for the same ice nucleation temperature over the entire year in Fig. 11. For example, Creamean et al. (2022, Fig. 4) observe constant INP values at T=-25°C over all seasons, whereas an annual cycle in the INP concentration is observed at T=-15°C. It would be interesting to see if the lidar can reproduce the in situ INP concentration over the entire year. Thus, I suggest to extrapolate the data shown in Fig. 11 to warmer/colder temperatures in winter/summer (e.g., show in situ and lidar INP concentrations at T=-25 °C and T=-15 °C over the entire year).

20. Fig. 11: Comments to Fig. 11:
    a. Why do you use the same temperature for 250 m and 2000 m in winter (-25 °C) but a different temperature in summer (-10 °C/-15 °C)?
    b. At which temperature were the in situ measurements conducted in winter? Please specify.
    c. Why are some symbols filled/empty? Please specify the color-coding.

21. Line 689-693: Here you state that the INP concentrations of 1-20 $L^{-1}$ are in consistency with MOSAiC retrievals of $n_{ICE}$ and that 10 MOSAiC cirrus systems were analyzed. To support this statement, I suggest to include the $n_{ICE}$ data points in Fig. 12.

**Technical comments**

1. Line 8: cloud condensation nucleus (CCN) and ice nucleating particle (INP) concentrations
2. Line 66: 'question' instead of 'questions'
3. Line 68-71: Long sentence. Please revise.
4. Line 72: 'Observations' instead of 'observation'
5. Line 108: The acronyms SML and BWS are introduced but not used in the paper
6. Line 109: 'midlatitudes' instead of 'midlatitudes latitudes'
7. Line 151: The acronym AOT was only introduced in the abstract. Should also be introduced in the main text.
8. Line 152: 'Sect. 3.2 and 3.3' instead of 'Sect. 3'
9. Line 158: 'between' instead of 'betwee'
10. Line 296: Add 'above to surface' to make it clear that the height levels of 250 m and 2000 m do not refer to 'below the tropopause'
11. Line 332: Replace one '$n_{INP}$' by '$n_{ICE}$'
12. Line 355: The acronym HULIS is introduced but not used in the paper
13. Line 369: Not sure if the acronym DIN is needed.
14. Line 436: Please reference to figure Fig. 1: '… above 1.5 km height (Fig. 1)'
15. Line 450: add reference to figure Fig. 1 '… was observed (Fig. 1)'.
16. Fig. 4 caption: 'smoke lidar ratio' instead of 'smome lidar ratio'
17. Line 475: add '… of the year-around backscatter observations …'

---

## Referee Comment (RC2)

**Review on the manuscript entitled "Annual cycle of aerosol properties over the central Arctic during MOSAiC 2019-2020 — light-extinction, CCN, and INP levels from the boundary layer to the tropopause" by Ansmann et al.**

This paper follows, as many others, the MOSAIC campaign conducted from the oceanographic vessel Polarstern that was conducted over the years 2019 and 2020. It is mainly focused on the exploitation of data from the Raman lidar that was on board the ship. It is a unique dataset from the ice pack because on an annual sampling of the Arctic atmosphere. The paper is very well organized and written. It contains many relevant references to support the statements. It is totally within the scientific domain of ACP.

For all these reasons, I think it can be published without major changes.

In addition, it raises current issues such as the impact of biogenic aerosols on ice cores or the role of biomass burning aerosol that can be mixed with terrigenous particles. These are still open scientific fields of importance for climate projections.

Below are some questions/comments.

Introduction.

1) It is very complete, maybe a little long. Some parts could be more synthesized, such as the discussions on CALIPSO that happen in two different places.

2) L16. I think we are talking about the middle troposphere?

3) L31. The ship was also trapped in the ice in August/September?

Section 2.
4) L212. The optical thicknesses are low, and this will induce very large errors on the Angstrom coefficient. It should also be taken into account in the interpretations.

5) In subsection 2.6, it is assumed that aerosols do not change in nature with altitude in order to apply the same coefficients c?

6) In fact, these coefficients c are the inverse of cross sections, why not use directly this very explicit quantity in physics?

7) L231. On which dataset is the regression done, it is not very clear?

8) L320. Can't there also be nitrates on the duts?

9) Subsection 2.8.3. If a thermodynamic model is used to calculate INPs from the lidar measurement, how can this independently validate the climate models? Don't these models use related approaches?

Section 3.
10) L464. How can we be sure that it is deep convection, linked to a pyroCb, that injects the wildfire smoke into the lower stratosphere? Is it the altitude at which the aerosols are observed by the lidar?

11) L467. We found the same thing on biomass burning aerosols from Canada (https://doi.org/10.5194/acp-18-13075-2018)

12) L475. In Fig. 6, how many lidar profiles are averaged per month? Are they homogeneously distributed in the month?

13) L544. Is it normal that the profiles of particle number concentration are not shown? It would have been interesting to see.

Section 4.

14) L591. In Fig. 10, the empty blue circles are not identified.

15) On Figures 10, 11 and 12, wouldn't it be clearer to put envelopes of data variation?

16) How can we separate natural variability and uncertainty from these figures?

17) L610-611. I don't quite understand the sentence about the dry deposit.

18) L631. Why did you take 1% at 250 m? Did the in-situ measurements give 1% dusts in number?

19) In-situ measurements are usually on mass, there must be significant errors to pass in numbers. Is this the case?

---

## Community Comment (CC1)

Regarding Fig.4, the optically dense upper tropospheric smoke layer and its attribution to a pyroCb in Canada or Alaska, the authors may benefit to learn that there were no pyroCbs detected in Canada at any time in 2020. There was a single pyroCb in Alaska in 2020, but in early June. There were pyroCbs in early September in California and Colorado, but they don't appear to be candidates for the Arctic smoke trajectories in Fig. 5. On 19 September, a smoke layer extremely similar to that in Fig. 4 was measured by the MOSAiC HSRL in northern Scandinavia (http://hsrl.ssec.wisc.edu/by_site/33/2020/09/19/am/#bscat_depol), and two days later by CALIOP at ~81N https://www-calipso.larc.nasa.gov/products/lidar/browse_images/show_v411_detail.php?s=production&v=V4-11&browse_date=2020-09-21&orbit_time=03-21-46&page=4&granule_name=CAL_LID_L1-Standard-V4-11.2020-09-21T03-21-46ZD.hdf. Back trajectories from these observations to 11-13 September suggest a connection with tropospheric wildfire smoke over the Pacific Ocean west of the USA. The Pacific plume episode is on display in this paper: https://acp.copernicus.org/articles/22/5399/2022/. If the back trajectories in Fig. 5 are run for a few more days, it is possible that some of them will curl in the direction of the Pacific smoke, which was not generated by pyroCbs. If these trajectories accurately connect the Polarstern smoke to its source, they are indicative of quasi-isentropic transport from the middle to upper troposphere.

**NOAA HYSPLIT MODEL**
**Backward trajectories ending at 0900 UTC 19 Sep 20**
**GFSQ Meteorological Data**

[Figure]

Job ID: 164523          Job Start: Sun May 14 11:55:31 UTC 2023
Source 1 lat.: 89.1  lon.: 110  hgts: 8500, 9000, 9500 m AMSL

Trajectory Direction: Backward     Duration: 168 hrs
Vertical Motion Calculation Method:     Model Vertical Velocity
Meteorology: 0000Z 19 Sep 2020 - GFS0p25

---

## Author Comment (AC1)

Dear reviewer,

thank you for careful reading of the manuscript and for providing valuable comments and ideas how to improve the paper.

A brief overview of main changes in the beginning:

We have now two separated sections, for MOSAiC instrumentation (Section 2) and for lidar retrievals (Section 3, aerosol microphysics, CCN, INP). Now the instrumental and data analysis methods are better separated. Furthermore, Section 3 is better structured with subsections from 3.1-3.3 and from 3.4.1-3.4.5.

We simplified the INP parameterization significantly. We have two INP retrieval sections now: Sect. 3.4.1. and Sect. 3.4.2. In Sect. 3.4.1., we describe the ABIFM approach (Knopf and Alpert, 2013) to estimate immersion freezing INPs from lidar observations in the lower troposphere. In Sect. 3.4.2, we describe the DIN approach (Wang and Knopf, 2011) to estimate deposition ice nucleation INPs from lidar observations in the upper troposphere. We removed the immersion freezing and deposition nucleation INP parameterizations of Ullrich et al. (2017). This was necessary in order to avoid 'mixing' of time-dependent and time-independent INP parameterizations…. and the difficulties, arising from this mixing of methods, when showing the obtained results in ONE MOSAiC time series. Now the full presentation of the annual cycle of INP observations is much more straight forward. We adjusted the discussion of the results after these changes.

We also changed the titles of Section 4 and 5 a bit: Section 4: Observations, part 1: …optical properties…, and Section 5: Observations, part2: cloud-relevant products, CCN, INP etc. This may better indicate that the RESULT block is separated into two main parts.

We introduced one new figure (Figure 12), as requested by one of the reviewers.

We went through the entire manuscript and improved the text after all the changes and with all the comments of the reviewers in mind.

Now the step-by-step response to all comments with our response in blue. Essential changes in the manuscript are indicated in BOLD.

This paper presents the annual cycle of height-resolved aerosol properties measured over the central Arctic by the MOSAiC lidar complemented with in situ observations. This unique dataset provides insights into the vertical distribution of optical and cloud-relevant aerosol properties (CCN, INP) throughout the troposphere and different seasons, which is important for modeling clouds and the present and future Arctic climate. The analysis of the annual cycle was extended with several case studies, with an emphasis on the presence of a persistent wildfire smoke layer in the upper troposphere.

The paper is well written and has a clear structure. I recommend publication after addressing the comments below.

General comments

1.   In Sect. 3.1 you present three case studies with clean and polluted conditions during summer before you discuss the annual cycle of the aerosol conditions during the MOSAiC year. The purpose of presenting these case studies is not yet clear to me. For example:
-   How did you select these case studies?
-       Is some information on the case studies required for the analysis of the annual cycle (e.g., derive lidar parameters that are used in the following sections)?

Please extend the description of the analysis regarding the case studies (e.g., goal of the analysis, selection of the case studies, …).

**In the beginning of Sect. 4.1, we explain: The three observations shown in Fig. 1 are selected because they cover the full range of MOASiC summer scenarios from clean to polluted conditions.**

**The case studies are not needed for the analysis of the annual cycle, but are useful for readers who are not just familiar with lidar profiling. It is a good tradition to start with some key days before one presents and discusses the entire observational data set in form of statistics and time series.**

2.  The detection of a pronounced and persistent wildfire smoke layer in the upper troposphere and lower stratosphere is an important MOSAiC highlight. However, I have the impression that the discussion about the smoke layer from previous studies was sometimes interrupting the flow. For example:
-       In line 474-475 you state that the annual cycle of the aerosol optical properties is shown in Fig. 6 and 7, so I was expecting that you would discuss the profiles of backscatter and extinction coefficients that you observed during MOSAiC. However, in the following lines you don't focus on the discussion of these figures/your observations, but you report the findings regarding the smoke layer from previous studies (see also specific comment 14).
-     In line 52-63 you put the discussion of the smoke layer together with the description of the instrument setup aboard Polarstern, which does not really fit.

I fully agree that the presence of the persistent wildfire smoke layer is an interesting feature/finding that needs to be discussed, but please revise if the structure of some paragraphs could be revised to maintain a clear flow.

**We agree with these comments and follow the suggestions of the reviewer and omitted almost completely the discussion on the UTLS wildfire smoke.**

Specific comments

1.  Line 113-118: This paragraph seems a bit out of place. You already introduced in the previous paragraph that local INPs of biogenic origin seem to control ice nucleation in the boundary layer in summer (line 101-102). Also, the definition of homogeneous and heterogeneous nucleation should be moved forward in the paper (before discussing the results of different INP studies). Please restructure the paragraphs regarding INPs. You could also add reference to Carlsen and David (2022) and Sze et al. (2023) as additional motivation for the importance of INP on cloud phase. For example, Carlsen and David (2022) saw a clear relationship between cloud phase over open ocean, snow and sea ice cover suggesting that local INPs are important for cloud phase.
Carlsen, T., & David, R. O. (2022). Spaceborne evidence that ice-nucleating particles influence high-latitude cloud phase. Geophysical Research Letters, 49, e2022GL098041. https://doi.org/10.1029/2022GL098041
Sze, K. C. H., Wex, H., Hartmann, M., Skov, H., Massling, A., Villanueva, D., and Stratmann, F.: Ice-nucleating particles in northern Greenland: annual cycles, biological contribution and parameterizations, Atmos. Chem. Phys., 23, 4741–4761, https://doi.org/10.5194/acp-23-4741-2023, 2023.

**We rearranged the second part of the Introduction and shortened it to make the entire argumentation in the Introduction more straight forward. Sze et al. (2023) and Carlsen and David (2022) are now considered in the motivating discussion in the Introduction.**

2.  Line 165-173: In subsection 2.2 you describe the lidar instrument. Is there a reason why you only

present the parameters measured/derived by the lidar in Sect. 2.5? I suggest presenting the parameters measured by the lidar already in Sect. 2.2. If you want to keep the parameters in a separate section, you should refer to section 2.5.

**We follow this suggestion, and present a short description of the lidar instrument together with the data analysis regarding directly measured quantities, i.e., backscatter, extinction, depol. ratio, and lidar ratio in Sect. 2.2.**

3.   Line 175: In the introduction you mention that sun photometer measurements are possible from March to September (line 88-89). Why were no measurements performed between March and June aboard Polarstern? I think it would be nice if you could include the sun photometer measurements in Fig. 9 (see specific comment 19).

**In Sect. 2.3, we state that the original MOSAiC sun photometer failed to work correctly. And we had no chance to bring a new one before June 2020.**
**We now present some photometer observations (mean AOD for June-July and August-September 2020) in the Fig. 9 caption. That should be sufficient.**

4.   Line 182-185: In Sect. 2.1 you state that Polarstern was mainly drifting at latitudes > 85 °N. The CALIOP measurements published by Yang et al. (2021) were obtained at latitudes between 65 °N and 82 °C (between 2006 and 2019). Thus, the CALIOP and MOSAiC measurements do not cover the same region and time period. How does this influence the comparison between MOSAiC and CALIOP measurements (e.g., Fig. 8 and 9)? Please add a few sentences regarding this point.

**We discuss this now in Sect. 4.2. The two main reasons for potential differences  are: CALIOP  sees more pollution from surrounding continents (65°-82°N) than the MOSAiC lidar (80°-90°N). Furthermore, CALIOP is looking downward and thus collects more tropospheric aerosol data  than the MOSAiC lidar that was frequently blocked by  low-level clouds and fog.**

5.   Table 1: What is the difference between the term 'exemplary' and 'typical' uncertainty? I would suggest sticking to one term.

**We now use 'typical'**

6.   Line 299-301: Please add a reference.

**This point (on most important INP types plus references)  is now discussed in larger detail in the Introduction.**

7.   Line 395-410: In Sect. 2.9 you describe the in situ measurements of aerosol microphysical properties and INP concentrations that were conducted on Polarstern. Maybe it makes sense to move Sect. 2.9 forward to the other instrumentation. Like that you first describe the applied instrumentation and then introduce the different data analysis methods.

**We did that, see new Sect. 2.5.**

8.   Line 396-410: Please include some information on how the particle number, CCN, INP concentrations were measured (e.g., instrument type) aboard Polarstern.

**Done, see Sect. 2.5!**

9.   Line 437: Where can I see the backward trajectories for the 4 km height? If they are not shown, please state that.

**Done in Sect. 4.1, on page 15!**

10. Fig 1: In the caption of Fig 1 you state that the observations with the near-range telescope are shown by the dashed line in Fig. 1.a. This is not the case. The near-range observations are represented by thinner lines. Please change the plot or caption accordingly.

**Done, i.e., we changed the caption!**

11. Fig. 3: Comments to Fig. 3:
a.   The 0 and 1 labels on the x-axis overlap. Please revise the labelling.
b.   For the right panel you show the residence time on the upper x-axis, while for the other panels you show the residence time on the lower x-axis. I suggest showing the residence time on the upper x-axis for all panels and the interval time on the bottom x-axis.

**We improved Fig. 3 accordingly.**

12. Line 475-486: The aim of this section is to present the annual cycle of the optical aerosol properties measured during the MOSAiC year (Fig. 6 and Fig. 7). However, in the first paragraph you focus mainly on the pronounced wildfire smoke layer that was discussed in detail by previous studies. I would suggest focusing on your observation and presenting Fig. 6 in detail before you discuss the results of other studies (see also general comment 2). For example, you could highlight the presence of the smoke layer in the upper troposphere (especially in winter) in Fig. 6.

**We re-wrote the entire Sect. 4.2 to meet all these points. We shortened the discussion on wildfire smoke in Sect. 4.2 drastically.**

13. Fig. 6: In the caption of Fig. 6 you state that 9-15 (October-March)/5-8 (June-September) daily observations per month were considered. Are those all measurements that were available or according to which criteria did you chose the observations? Please specify. How many observations were considered for April and May? Please add the number of observations also for April and May. Alternatively, you can specify the number of observations for each one- month/two-month period in brackets in the legend of the figure.

**We clarify all this is in the first paragraph of Sect. 4.2. We provide total numbers of considered observations per month for each of the 12 MOSAiC months.**

14. Line 532-533: Here you state that the MOSAiC and CALIOP observations agree well during the winter months (Fig. 8). However, above 7.5 km, the extinction coefficients measured during MOSAiC are higher compared to the CALIOP measurements, which is an effect of the smoke layer. You could include this in the interpretation of the figure. The same holds true for the extinction coefficients measured in the upper troposphere during the summer season.

**We re-phrased this part of the CALIOP-MOSAiC comparison to meet the reviewer's point.**

15. Fig. 8: Comments to Fig. 8:
a.   I think it would be beneficial if you could include the standard deviation for the CALIOP measurements. This allows to assess if the MOSAiC year was significantly different compared to the 15-year mean CALIOP profile.

**In the Yang et al. (2021) paper, there are no standard deviations given. In order to provide some information about variability, we now show the standard deviations for our MOSAiC observations.**

b. To support the comparison between the 15-year CALIOP and MOSAiC lidar measurements, you could investigate if the MOSAiC and CALIOP measurements of the same time period are in good agreement: I.e. can the CALIOP profile from October
2019 to September 2020 capture the vertical profile measured by MOSAiC?

**All this information is not available in the Yang et al. (2021) paper. We could do our own CALIOP data analysis to cover exclusively the MOSAiC year. But that would be quite time consuming. Since the CALIOP-MOSAiC comparison is not the main goal of the article, we did not follow this reasonable suggestion.**

16. Line 537-549: In this paragraph you compare height-resolved aerosol observations from MOSAiC, a TBS and CALIOP. However, these observations were obtained in different regions and over different time periods, which makes a direct comparison difficult. What is the goal of this comparison? In my opinion it does not make sense to compare the aerosol observations of these three datasets, as they are quite different. As this paragraph is not required for the outline of the paper, I suggest removing this paragraph. If you decide to keep this comparison in the final manuscript, you should include a figure to illustrate the comparison. Also, I am a bit surprised that you state that good agreement between the MOSAiC, CALIOP and TBS aerosol profiles was found (line 544-545), when you write in line 533-534 that in summer, the lower troposphere measured during MOSAiC was much cleaner than described by the 15-year mean CALIOP profiles.

**We followed this suggestion and removed this part completely.**

17. Fig. 9: Comments to Fig. 9:
a. In Sect. 2.3 you introduce the sun photometer measurements and state that these measurements were performed between June to September. I think it would be beneficial to include the sun photometer measurements in Fig. 9.
b. Please include the definition of the vertical bar in the caption (uncertainty?). Why is there no vertical range for AOT 0-20 km? Why are there no data points between May and September for AOT 0-20 km? Please revise.

**We improved Fig. 9, and now show standard deviation (SB bars), indicating the natural variability for all three MOSAiC time series and mention that in the caption. In addition, we include mean AOT values (sun photomter, June-July, August-September 2020) in the caption, not in the figure.**
**In Sect. 4.3, the 0-20km AOT stops in April 2020 in Fig. 9 because a clear difference between the 0-20km and 0-10km AOT was no longer visible in the data.**
**This is mentioned in Sect. 4.3.**

18. Fig. 10: Comments to Fig. 10:
a. The lower standard deviation bar is missing for some values. Please include it. b. What is the difference between the filled/empty blue circle? Please specify.

**We changed (simplified) the entire data analysis. We used different conversion factors for summer and winter, now we use the Arctic aerosol conversion factors throughout the year, and therefore we have only closed symbols (in the case of the lidar observations).**

**In the case of the in situ observations, the lower value of the SD bar is sometimes <1 or even partly <0. Therefore, this lower end of the bar is not shown. This is mentioned in the caption now.**

19. Line 637-650: In this section you describe the strong difference between the winter and summer INP levels that is visible in Fig. 11. This pronounced annual cycle in the INP concentration can be mainly explained by the difference in summer/winter temperatures, as ice nucleation is strongly temperature dependent. To investigate the annual cycle of the INP concentration you should show the INP

concentration for the same ice nucleation temperature over the entire year in Fig. 11. For example, Creamean et al. (2022, Fig. 4) observe constant INP values at T=-25°C over all seasons, whereas an annual cycle in the INP concentration is observed at T=-15°C. It would be interesting to see if the lidar can reproduce the in situ INP concentration over the entire year. Thus, I suggest to extrapolate the data shown in Fig. 11 to warmer/colder temperatures in winter/summer (e.g., show in situ and lidar INP concentrations at T=-25 °C and T=-15 °C over the entire year).

**All this is now done (as suggested by the reviewer), see section 5.2. We introduced a new Figure 12 (INP concentrations for T=-25 °C and T=-15 °C). We went through the entire text of Section 5.2 to improve the discussion, after integrating Fig.12, along the reviewers suggestions.**

20. Fig. 11: Comments to Fig. 11:
a. Why do you use the same temperature for 250 m and 2000 m in winter (-25 °C) but a different temperature in summer (-10 °C/-15 °C)?
b. At which temperature were the in situ measurements conducted in winter? Please specify.
c. Why are some symbols filled/empty? Please specify the color-coding.

**We improved the discussion and better explain Fig. 11. We used the temperatures of -25°C and -10° to -15°C as realistic cloud top temperatures in winter and summer, respectively. We used these different temperatures just to show how strong the differences in INP concentrations from winter to summer are.**
**We provide now the mean INP values (and SD) for winter and summer seasons for surface, 250 m, 2000 m in the text, addition, as in the case of CCN discussions.**

21. Line 689-693: Here you state that the INP concentrations of 1-20 $L^{-1}$ are in consistency with MOSAiC retrievals of $n_{ICE}$ and that 10 MOSAiC cirrus systems were analyzed. To support this statement, I suggest to include the $n_{ICE}$ data points in Fig. 12.

**We added a paragraph on this on page 24 in Sect. 5.3 (use of CAPTIVATE method, Mason et al., 2023) and provide some results. Meanwhile we analyzed around 10 long-lasting cirrus events (from the birth of the cirrus to the dissolution of the cirrus system) in November, December, January and February of the MOSAiC year. But these results are still preliminary and are presently checked and compared with results by using an alternative approach to obtain proper numbers for the uncertainties in these ICNC retrievals. But these CALTIVATE results are at all in line with the INPC values of 1-20 per liter. In most cases, the CAPTIVATE results indicate even lower ice crystal number concentrations, 0.1-1 crystals per liter….**

Technical comments

1. Line 8: cloud condensation nucleus (CCN) and ice nucleating particle (INP) concentrations

**Done**

2. Line 66: 'question' instead of 'questions'

**Done**

3. Line 68-71: Long sentence. Please revise.

**We re-phrased this part in the Introduticon.**

4. Line 72: 'Observations' instead of 'observation'   **ok!**

5. Line 108: The acronyms SML and BWS are introduced but not used in the paper

**They are removed.**

6. Line 109: 'midlatitudes' instead of 'midlatitudes latitudes'

**ok!**

7. Line 151: The acronym AOT was only introduced in the abstract. Should also be introduced in the main text.

**Improved!**

8. Line 152: 'Sect. 3.2 and 3.3' instead of 'Sect. 3'

**Improved!**

9. Line 158: 'between' instead of 'betwee'

**Improved!**

10. Line 296: Add 'above to surface' to make it clear that the height levels of 250 m and 2000 m do not refer to 'below the tropopause'

**Improved!**

11. Line 332: Replace one 'nINP' by 'nICE'

**Improved!**

12. Line 355: The acronym HULIS is introduced but not used in the paper

**We removed HULIS.**

13. Line 369: Not sure if the acronym DIN is needed.

**DIN is needed!**

14. Line 436: Please reference to figure Fig. 1:  '… above  1.5 km height (Fig. 1)'

**Done!**

15. Line 450: add reference to figure Fig. 1 '… was observed (Fig. 1)'.

**Done!**

16. Fig. 4 caption: 'smoke lidar ratio' instead of 'smome lidar ratio'

**Done!**

17. Line 475: add '… of the year-around backscatter observations …'

**Improved!**

---

## Author Comment (AC2)

Dear reviewer,

thank you for careful reading of the manuscript and for providing valuable comments and ideas how to improve the paper.

A brief overview of main changes in the beginning:

We have now two separated sections, for MOSAiC instrumentation (Section 2) and for lidar retrievals (Section 3, aerosol microphysics, CCN, INP).  Now the instrumental and data analysis methods are better separated. Furthermore, Section 3 is better structured with subsections from 3.1-3.3 and from 3.4.1-3.4.5.

We simplified the INP parameterization significantly. We have two INP retrieval sections now: Sect. 3.4.1. and Sect. 3.4.2. In Sect. 3.4.1., we describe the ABIFM approach (Knopf and Alpert, 2013) to estimate immersion freezing INPs from lidar observations in the lower troposphere. In Sect. 3.4.2, we describe the DIN approach (Wang and Knopf, 2011) to estimate deposition ice nucleation INPs from lidar observations in the upper troposphere. We removed the immersion freezing and deposition nucleation INP parameterizations of Ullrich et al. (2017). This was necessary in order to avoid 'mixing' of time-dependent and time-independent INP parameterizations…. and the difficulties, arising from this mixing of methods, when showing the obtained results in ONE MOSAiC time series. Now the full presentation of the annual cycle of INP observations is much more straight forward. We adjusted the discussion of the results after these changes.

We also changed the titles of Section 4 and 5 a bit:  Section 4: Observations, part 1: …optical properties…, and  Section 5: Observations, part2: cloud-relevant products, CCN, INP etc. This may better indicate that the RESULT block is separated into two main parts.

We introduced one new figure (Figure 12), as requested by one of the reviewers.

We went through the entire manuscript and improved the text after all the changes and with all the comments of the reviewers in mind.

Now the  step-by-step response to all comments with our response in blue. Essential changes in the manuscript are indicated in BOLD.

This paper follows, as many others, the MOSAiC campaign conducted from the oceanographic vessel Polarstern that was conducted over the years 2019 and 2020. It is mainly focused on the exploitation of data from the Raman lidar that was on board the ship. It is a unique dataset from the ice pack because on an annual sampling of the Arctic atmosphere. The paper is very well organized and written. It contains many relevant references to support the statements. It is totally within the scientific domain of ACP.

For all these reasons, I think it can be published without major changes.

In addition, it raises current issues such as the impact of biogenic aerosols on ice cores or the role of biomass burning aerosol that can be mixed with terrigenous particles. These are still open scientific fields of importance for climate projections.

**This comment motivated us to even include recent results of Tobo et al. (2019) and Kawai et al. (2023) who pointed to the importance of high latitude dust (glacier washout products containing biogenic material) regarding heterogeneous ice nucleation at high temperatures, >-15°C (on page 3).**

Below are some questions/comments.

Introduction.

1) It is very complete, maybe a little long. Some parts could be more synthesized, such as the discussions on CALIPSO that happen in two different places.

**We agree and shortened the Introduction considerably (submitted version: 3.5 pages, revised version: 2 pages).**

2) L16. I think we are talking about the middle troposphere?

**In the Abstract, we now write: … caused a re-increase of the aerosol concentration towards the tropopause…..**

3) L31. The ship was also trapped in the ice in August/September?
**Polarstern was again at latitudes >85°N from 21 August to 20 September 2020 and thus in the ice. This is now written on page 3 in the Introduction.**

Section 2.

4) L212. The optical thicknesses are low, and this will induce very large errors on the Angstrom coefficient. It should also be taken into account in the interpretations.

**Yes, we agree, we re-phrased the text a bit. However, all the AERONET observations in the Arctic shows such a narrow range of Angstrom values. We think, all these Angstrom values are trustworthy.**

5) In subsection 2.6, it is assumed that aerosols do not change in nature with altitude in order to apply the same coefficients c?

**We mention now at several places in the revised text that we assume similar aerosol conditions up to 3 km height, so that we can apply the conversion factors for the 250 and 2000 m height levels.**
**We also discuss that these conversion factors (based on summer time AERONET observations) may not be fully appropriate for Arctic haze conditions (winter time aerosol conditions).**

6) In fact, these coefficients c are the inverse of cross sections, why not use directly this very explicit quantity in physics?

**You are right! On the other hand, we have these simple formulas, Eqs.(1)-(3), in Sect. 3.1, we leave it as is…. in order to confuse readers not too much.**

7) L231. On which dataset is the regression done, it is not very clear?

**We explain all this now in more detail in Sect. 3.1 (in the paragraph after introducing Eqs.(1)-(3)). The full procedure is, however, described in Mamouri and Ansmann (2016). That is mentioned as well.**

8) L320. Can't there also be nitrates on the dust?

**Yes! By passing through polluted regions dust particles can become coated with sulfate, nitrate, and organic substances. This is now mentioned on page 11 in Sect. 3.4.1.**

9) Subsection 2.8.3. If a thermodynamic model is used to calculate INPs from the lidar measurement, how can this independently validate the climate models? Don't these models use related approaches?

**The INP parameterizations are obtained from laboratory studies, i.e., from observations at well-defined temperature and humidity conditions and for well-defined aerosol properties**

**(chemical composition, size distribution). They are not obtained from modelling. In the lidar retrieval we us temperature, humidity, and aerosol information to estimate INP concentrations by using the laboratory findings. Climate modelers may use the same INP parameterizations.**

**We do not want to validate climate models?**

Section 3.

10) L464. How can we be sure that it is deep convection, linked to a pyroCb, that injects the wildfire smoke into the lower stratosphere? Is it the altitude at which the aerosols are observed by the lidar?

**No, we cannot be sure! Mike Fromm (reviewer #3) had a similar comment. Now we provide an extended discussion on the potential smoke sources in the last paragraph in Sect. 4.1. We include the findings in the article of Hu et al. (2022). These authors observed the Californian smoke (that partly travelled to the Arctic from 10-19 September 2020).**

**According to Hu et al. (2022}, intensive wildfires in California and Oregon injected large amounts of wildfire smoke into the atmosphere on 10 and 11 September 2020. Thick smoke layers at 5-10km height were detected with CALIOP over the Pacific Ocean just west of the west coast of North America Hu et al. (2022} ……….. Hu et al. (2022) mentioned that pyrocumulonimbus (pyroCb) development occurred on 9 September and that the smoke was trapped over the eastern Pacific Ocean on 7-11 September due to cyclone activity. It remains open to what extent strong convective motions were responsible for smoke lofting up to the upper troposphere.**

11) L467. We found the same thing on biomass burning aerosols from Canada (https://doi.org/10.5194/acp-18-13075-2018)#

**We now consider this (plus reference) in the last paragraph in Sect. 4.1.**

12) L475. In Fig. 6, how many lidar profiles are averaged per month? Are they homogeneously distributed in the month?

**No! They are not homogeneously distributed. We used all the cloudfree periods for aerosol profiling. Now, we provide numbers of observations per months for all MOSAiC months (Sect. 4.2, first paragraph).**

13) L544. Is it normal that the profiles of particle number concentration are not shown? It would have been interesting to see.

**The number concentrations are the most uncertain products (surface area and volume concentrations are much more robust retrieval products). We discuss that a bit more at several places (Sects.3 and 5). This is the main reason why we hesitate to show number concentration profiles.**

Section 4.

14) L591. In Fig. 10, the empty blue circles are not identified.

**We use one set of conversion factors (for Arctic summer aerosol). In the submitted version, we used conversion factors for marine aerosol, i.e., for sea salt, in summer (Mamouri and Ansmann, 2016). That was a mistake! We changed that and use the conversion factors for Arctic summer aerosol throughout the MOSAiC year now. Therefore, we have only closed circles in Fig.10 of the revised version.**

15) On Figures 10, 11 and 12, wouldn't it be clearer to put envelopes of data variation?

**We do not like this idea! However, we now provide seasonal mean CCN values (for winter and summer seasons, for surface, 250, and 2000 m height) in the text, in Sect. 5.1. This helps to see the variations in the data, and these variations are mostly due to natural (atmospheric) variations.**

16) How can we separate natural variability and uncertainty from these figures?

**Table 1 contains uncertainty information for the lidar products. The uncertainty in the in situ observations is lower. We now provide mean values and standard deviations of measured data, and the obtained variability of 50-100% can be interpreted as natural or atmospheric variability. That is at least our position.**

17) L610-611. I don't quite understand the sentence about the dry deposit.

**We removed the paragraph on low level jets to keep the discussion short and to avoid confusion by too many hypotheses.**

18) L631. Why did you take 1% at 250 m? Did the in-situ measurements give 1% dusts in number?

**We adjusted our lidar derived INP values to the in situ measured INP values. This is now described in Sect. 5.2 (page 21).**

19) In-situ measurements are usually on mass, there must be significant errors to pass in numbers. Is this the case?

**In Sect. 2.5, we now provide details on the methods regarding the in situ observations aboard Polarstern. Particle number concentrations and INP number concentrations are derived from these observations. The uncertainty analysis is described in the respective MOSAiC papers of Boyer et al. (2022) and Creamean et al. (2022). Well-characterized and well-established retrieval methods are used. The uncertainties are certainly much lower than 50%.**

---

## Author Comment (AC3)

**Dear Mike Fromm,**

**thank you for your comment. Our answer in blue.**

Regarding Fig.4, the optically dense upper tropospheric smoke layer and its attribution to a pyroCb in Canada or Alaska, the authors may benefit to learn that there were no pyroCbs detected in Canada at any time in 2020. There was a single pyroCb in Alaska in 2020, but in early June. There were pyroCbs in early September in California and Colorado, but they don't appear to be candidates for the Arctic smoke trajectories in Fig.5. On 19 September, a smoke layer extremely similar to that in Fig. 4 was measured by the MOSAiC HSRL in northern Scandinavia (http://hsrl.ssec.wisc.edu/by_site/33/2020/09/19/am/#bscat_depol), and two days later by CALIOP at ~81N https://www- calipso.larc.nasa.gov/products/lidar/browse_images/show_v41 1_detail.php?s=production&v=V4-11&browse_date=2020-09- 21&orbit_time=03-21-46&page=4&granule_name=CAL_LID_L1- Standard-V4-11.2020-09-21T03-21- 46ZD.hdf. Back trajectories from these observations to 11-13 September suggest a connection with tropospheric wildfire smoke over the Pacific Ocean west of the USA. The Pacific plume episode is on display in this paper: https://acp.copernicus.org/articles/22/5399/2022/. If the back trajectories in Fig. 5 are run for a few more days, it is possible that some of them will curl in the direction of the Pacific smoke, which was not generated by pyroCbs. If these trajectories accurately connect the Polarstern smoke to its source, they are indicative of quasi-isentropic transport from the middle to upper troposphere.

[Figure]

**Motivated by the comment of Mike Fromm, we studied the paper of Hu et al. (2022), and extended the discussion on the smoke source region and the potential impact of pyroCbs.**

**In Sect. 4.1, pages 15-16, we write: According to Hu et al. (2022}, intensive wildfires in California and Oregon injected large amounts of wildfire smoke into the atmosphere on 10 and 11 September 2020. Thick smoke layers at 5-10 km height were detected with CALIOP over the Pacific Ocean just west of the west coast of North America (Hu et al., 2022).**

**.....**

**Hu et al. (2022) mentioned that pyrocumulonimbus (pyroCb) development occurred on 9 September and that the smoke was trapped over the eastern Pacific Ocean on 7-11 September due to cyclone activity. It remains open to what extent strong convective motions were responsible for smoke lofting up to the upper troposphere.**